

# Data-driven farm-wide fatigue estimation on jacket foundation OWTs for multiple SHM setups

Francisco d N Santos[1], Nymfa Noppe[1], Wout Weijtjens[1], and Christof Devriendt[1]

[1]OWI-Lab, Vrije Universiteit Brussel, Pleinlaan 2, 1050 Brussels

**Correspondence:** Francisco de N Santos (francisco.de.nolasco.santos@vub.be)

**Abstract.**

The sustained development over the past decades of the offshore wind industry has seen older wind farms beginning to reach their design lifetime. This has led to a greater interest in wind turbine fatigue, the remaining useful lifetime and lifetime extensions. In an attempt to quantify the progression of fatigue life for offshore wind turbines, also referred to as a fatigue assessment, structural health monitoring (SHM) appears as a valuable contribution. Accurate information from a SHM system, can enable informed decisions regarding lifetime extensions. Unfortunately direct measurement of fatigue loads typically revolves around the use of strain gauges and the installation of strain gauges on all turbines of a given farm is generally not considered economically feasible. However, when we consider that great amounts of data, such as Supervisory Control And Data Acquisition (SCADA) and accelerometer data (of cheaper installation than strain gauges), is already being captured, this data might be used to circumvent the lack of direct measurements.

It is then highly relevant to know what is the minimal sensor instrumentation required for a proper fatigue assessment. In order to determine this minimal instrumentation, a data-driven methodology is developed for real-world jacket-foundation Offshore Wind Turbines (OWT). Firstly, high-frequent 1s SCADA data is used to train an Artificial Neural Network (ANN) that seeks to estimate the quasi-static thrust load, and able to accurately estimate the thrust load with a Mean Absolute Error (MAE) below 2 %. The thrust load is then, along with 1s SCADA and acceleration data, processed into 10-minute metrics and undergoes a comparative analysis of feature selection algorithms with the goal of performing the most efficient dimensionality reduction possible. The features selected by each method are compared and related to the sensors. The variables chosen by the best-performing feature selection algorithm then serve as the input for a second ANN which estimates the tower fore-aft (FA) bending moment Damage Equivalent Loads (DEL), a valuable metric closely related to fatigue. This approach can then be understood as a two-tier model: the first tier concerns itself with engineering and processing 10 minute features, which will serve as an input for the second tier.

It is this two-tier methodology that is used to assess the performance of 8 realistic instrumentation setups (ranging from 10 minute SCADA to 1s SCADA, thrust load and dedicated tower SHM accelerometers). Amongst other findings, it was seen that accelerations are essential for the model's generalization. The best performing instrumentation setup is looked in greater depth, with validation results of the tower FA DEL ANN model show an accuracy of around 1 % (MAE) for the training turbine and below 3 % for other turbines, with a slight underprediction of fatigue rates. Finally, the ANN DEL estimation model - based on





a intermediate instrumentation setup (1s SCADA, thrust load, low quality accelerations) - is employed in a farm-wide setting, and the probable causes for outlier behaviour investigated.

# 1 Introduction

## 1.1 Fatigue assessment

Topics such as the fatigue experienced by offshore wind turbines, their remaining useful lifetime and foreseeable lifetime extensions have become increasingly crucial topics for the offshore wind energy sector, particularly as older wind farms begin to reach the end of their design lifetime. Taking into account the fatigue assessment of turbines is crucial if operators are to make informed decisions regarding wind turbine's lifetime extension. Collecting data required for such fatigue assessments is generally considered a part of Structural Health Monitoring (SHM). Martinez-Luengo and Shafiee (2019) has shown how, although initially increasing the capital expenditures as some additional hardware is required, SHM induces a reduction in operational expenditure which far exceeds the initial increase in capital expenditures. Thus, SHM is highly attractive in the current industry climate, as it allows to reduce overall costs which can then be translated into a further reduction of the Cost of Energy (CoE), one of the main challenges of the industry at large (Van Kuik et al., 2016). Furthermore, offshore wind turbine design is usually driven by fatigue, wherein improvements in fatigue assessment of built wind turbines can induce further optimization of future designs (Seidel et al., 2016).

Fatigue assessments are often based on measurements of the turbine's load history (Loraux and Brühwiler, 2016; Schedat et al., 2016; Iliopoulos et al., 2017; Ziegler et al., 2017). A direct measurement of fatigue loads is obtained through the use of strain gauges. Strain gauges on the substructure's primary steel allow to measure the strain histories. These strain histories can then be readily translated to stress histories and ultimately fatigue loads, e.g. through the use of rainflow counting. Unfortunately, the installation and operation of strain gauges is rather labour and maintenance intensive, resulting in a rather limited industry-adoption. At best only a subset of turbines in a farm are equipped with strain gauges to monitor the fatigue life of the substructure. In contrast operators do want to understand the fatigue rates across the entire wind farm and as such alternatives to quantify fatigue loads are being searched. In particular the use of supervisory control and data acquisition (SCADA) data is often considered. SCADA data is interesting as it captures the key operational data (e.g. power production, wind speed, blade pitch, ...) of an (offshore) wind turbine. SCADA is also available for every turbine and is stored by most operators. When a method can be developed that estimates fatigue rates from SCADA data, then farm-wide fatigue rates can be obtained.

However, for offshore wind turbines the sole use of SCADA data might be insufficient to fully explain fatigue behavior. While the turbine control and environmental conditions cover a significant part of fatigue loads, the interaction between dynamic loads and the structural dynamics of the substructure also play a key role (Vorpahl et al., 2013). It is this interaction that is typically poorly represented by the SCADA data[1]. In contrast, accelerometer data almost exclusively covers those structural

---

[1]In this contribution, for clarity, we don't consider any accelerometer data to be part of the SCADA data of the wind turbine and will refer to it separately. Even though for some operators accelerations are collected as part of the SCADA.





dynamics and is closely related to the stress histories. We can see how closely related the accelerometer data is to the strain measurements in Figure 1. Therefore the inclusion of accelerometer data is considered to complement SCADA data.

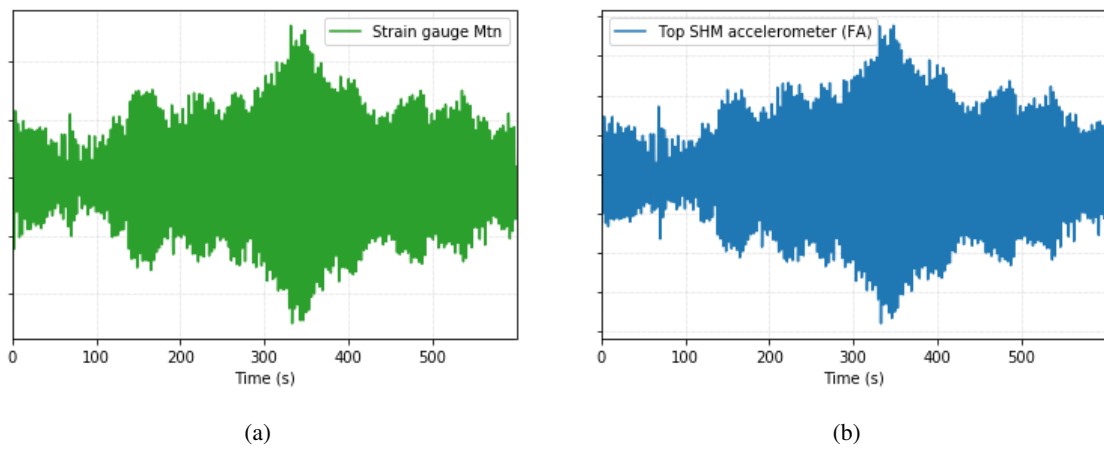

(a)                                                                                    (b)

**Figure 1.** (a) Fore-aft bending moment ($M_{tn}$, $Nm$) signal, measured by the strain gauges during a standstill occurrence of $600s$. (b) Top tower SHM accelerometer fore-aft displacement signal ($m$) for the same period as in (a).

In this contribution, we investigate a solution to estimate the fatigue rates (expressed as Damage Equivalent Loads, DEL) for an offshore wind farm on jacket foundations. At this farm high-frequent 1s SCADA and (low-quality) nacelle-installed accelerometers are available for all locations. At two locations a dedicated SHM system is installed comprising tower accelerometers and strain gauges.

The aim is to predict the 10 minute fatigue rates (DEL) of the entire farm. This is achieved by training an Artificial Neural
Network (ANN) using the data from one of the two SHM-locations, to predict the DEL solely using 10 minute statistics derived from the 1s SCADA and accelerometer data. All data is brought back to 10 minute intervals as this is the lowest *common denominator* of most sensors' measurement intervals. For various parameters, including DEL, only values at 10 minute intervals are available. Moreover working with 10 minute intervals, over 1s intervals, dramatically reduces the amount of data required to represent long periods of time (e.g. several years) and avoids issues with time-synchronization between various systems that
can easily accumulate to a difference of several seconds.

In this contribution we had access to the 1s SCADA, allowing to assess the added value of various statistics at a ten minute-interval. Unique to the current investigation is the use of a second ANN that translates 1s SCADA into an estimate of the thrust load at $1Hz$. This offers a direct look into the loads of the turbine even in absence of strain gauges. This thrust load estimate is then, much like all other 1s SCADA parameters, translated in 10minute interval metrics using various statistics. As an ANN is
used both on the 1s SCADA and in the prediction of the 10-minute DELs we refer to our approach as Two Tier.

This contribution aims to assess the feasibility of this strategy, study the added value of various sensors and statistics and provide insight in how the most suitable parameters can be selected.





## 1.2 Use of machine learning in (offshore) wind

The increasing adoption of data acquisition systems in modern wind turbines and the large amount of data they produce,
combined with the advent of widespread use of artificial intelligence (AI), has led to an increased use of ANNs within the
specific context of wind energy, exhaustively documented by Marugán et al. (2018), Wilkinson et al. (2014) and Stetco et al.
(2019), with the latter having a clear focus on condition monitoring (Helsen et al., 2015). Thus, data-driven approaches present
themselves as increasingly alluring alternatives to physics-based models, assuming themselves as the next step in operational
fatigue lifetime estimation (Veldkamp, 2008).

Data-driven approaches appear then to be especially suitable to predict tower fore-aft (FA) bending moment damage equiva-
lent loads (DEL). Previous research has often shown the high sensitivity of neural networks to input variables' quality (Novak
et al., 2018), which renders proper selection of input variables paramount to the model's performance (Leray and Gallinari,
1999), consisting in a good practice to uphold (Vera-Tudela and Kühn, 2014). To this point, an input feature engineering and
selection methodology was developed based on 10-minute metrics of several input parameters from SCADA, accelerations and
thrust load data. The results of the input feature selection are thoroughly analysed, and their validity and applicability discussed
in the present contribution.

Preceding relevant research from Smolka et al. (2013), has successfully investigated the possibility of establishing a reliable
data-driven fatigue estimator serving the entirety of the turbine's operational life and what amount and type of sample data is
required, provided an accurate portrayal of the diversity of loading situations and rigorous sample selection are present.

Likewise, the work of Vera-Tudela and Kühn (Vera-Tudela and Kühn, 2014, 2017), albeit dealing with the blade flap- and
edgewise bending moment fatigue load estimation, has put forth a robust methodology to evaluate the accuracy of different
feature selection methods and used one year of measurements at two wind turbines to evaluate the prediction quality of their
SCADA-based neural network model in different flow conditions with acceptable results.

Similarly, Avendaño-Valencia et al. (2021) has used Gaussian Process Regression time-series modelling to evaluate the
influence so-called EOPs (Environmental and Operational Parameters) have on the features of the vibration response of the
wind turbine blades. Also applied to estimate the blade root flapwise damage equivalent loads (DEL), Schröder's work has
emphatically demonstrated how a surrogate model based on ANNs outperforms other surrogate models, such as polynomial
chaos expansion and quadratic response surface, in computational time, model accuracy and robustness, further applying it to
connect wind farm loads to turbine failures (Schröder, 2020).

Finally, Movsessian et al. (2021) has used one-year SCADA data from onshore wind turbines to perform a comparative
analysis of feature selection techniques, in particular assessing the strengths of Neighbourhood Component Analysis (NCA)
(Goldberger et al., 2004) when compared to other feature selection algorithms, such as Pearson's correlation, Principal Com-
ponent Analysis (PCA) (Wold et al., 1987) and stepwise regression (Draper and Smith, 1998), to estimate the tower fore-aft
bending moment.





## 2 Sensors, Data and Methodology

### 2.1 Sensors and Data

The current contribution is part of a long-standing effort where OWI-Lab aims to develop structural health monitoring procedures based on load and vibration measurements, which enable more accurate lifetime predictions in Offshore Wind Turbines (OWT). It is in this context that data is acquired from two real-world instrumented OWTs on jacket foundations within the an offshore wind farm. An overview on the placement and data collected by the different sensors of the fully-instrumented turbines can be seen in the schematic presented in Figure 2.

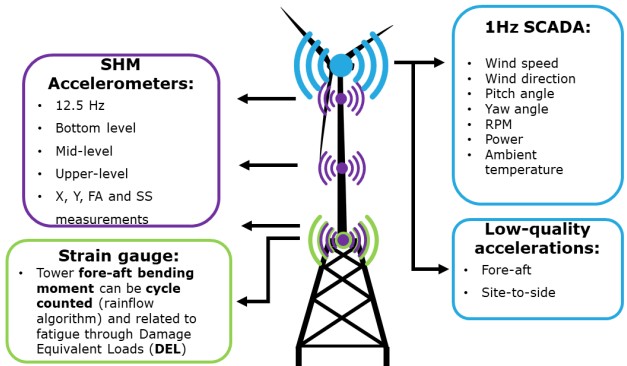

**Figure 2.** Overview of different sensors installed in wind turbine. Sensors locations are not accurate representations.

For this particular farm SCADA data at a $1Hz$ sample frequency was available for every turbine. The 1s SCADA data contained, a.o. wind speed ($m/s$), wind direction (°), pitch angle (°), yaw angle (°), rotational speed/RPM (cps), power ($kW$) and ambient temperature (°). This wind farm also collected acceleration data from a built-in bi-axial accelerometer in the nacelle on every turbine. However, this sensor was like the SCADA data sampled at a frequency of $1Hz$.

In addition data from two turbines with a full SHM setup was available. The SHM setup comprises three dedicated tower bi-axial accelerometers with a sampling frequency of 12.5 $Hz$ located at the tower bottom-, mid- and top-level. Apart from the accelerations in the two sensor directions, also the accelerations in the nacelle's frame of reference (i.e. fore-aft (FA) and side-to-side (SS)) are calculated through the use of the known yaw angle of the wind turbine. In addition the collected SHM accelerations can also be transformed into displacements, through double integration in the frequency domain. To avoid excessive drifts in this transformation a lower frequency bound of $0.1Hz$ is used.

These accelerometers outperform the accelerometers in the nacelle in three key properties: firstly the higher sampling frequency offers a wider frequency range, secondly they offered a much better signal to noise ratio than those installed in the nacelle. A third disadvantage of the nacelle accelerometer was that only the absolute values of this sensor were stored. A timeseries of the nacelle accelerometer is shown on top of one from the SHM system in Figure 3.



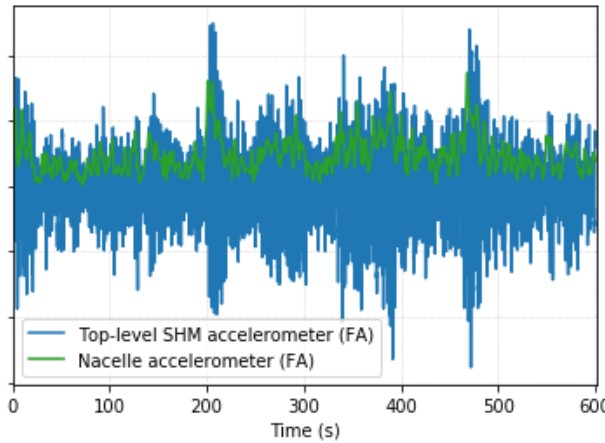

**Figure 3.** Timeseries comparing the top-level FA SHM accelerometer signal and the nacelle-installed FA signal.

Alongside the accelerometers, the SHM setup also contains 4 axial strain gauges installed along the transition piece (TP) inner circumference at the TP-Tower interface level. This setup of 4 strain gauges allows to calculate the bending moments in both FA ($M_{tn}$) and SS ($M_{tl}$) direction, when the yaw angle is known (Link and Weiland, 2014). For fatigue assessment the setup also cycle-counts all strain and bending moment histories and calculates damage equivalent loads (DEL). After obtaining the bending moments, one can then employ a rainflow counting algorithm (Dirlik, 1985) (reliant on a subset of the python implementation of the WAFO toolbox (Brodtkorb et al., 2000)), a well-known method widely discussed in literature (Marsh et al., 2016), which combines the number of cycles with their stress range. Finally, through the employment of a SN curve (Ziegler and Muskulus, 2016) and Palmgren-Miner's rule (Kauzlarich, 1989), the damage equivalent loads are calculated. A more detailed discussion of this procedure can be found in Hübler et al. (2018).

Damage equivalent loads are usually presented under two forms: damage equivalent moments (DEM) or damage equivalent stress ranges (DES). These do not present two different quantities, but rather, two ways of presenting the same information (respectively as bending moments ($Nm$) or as stress $MPa$), easily translatable between using the moment of inertia. We can then understand the DEL as an umbrella-term for both DEM and DES. The DEM is based on Eq. (1) as defined by Hendriks and Bulder (1995) (here presented for the stress ranges), wherein $m$ is the slope of the S-N curve, $n_i$ is the number of cycles of a given stress range, $\sigma_i$, $r_o$, the tower outer radius, $r_i$, the tower inner radius and $N_{eq} = 10^7$, a predefined number of cycles. Following the discussion in Seidel et al. (2016), the compromise value of 4 was selected for $m$.

$$DEM = \left( \frac{\sum_i n_i \cdot \left( \frac{\Delta\sigma_i \cdot \frac{\pi}{2} \cdot (r_o^4 - r_i^4)}{r_i} \right)^m}{N_{eq}} \right)^{1/m} \tag{1}$$

As the DEL is a direct quantification of fatigue loads it can be considered as the primary input for any future fatigue assessment. But naturally these DEL values are only available for the two instrumented turbines. This paper aims to determine





a methodology to estimate these DEL values for all turbines in the farm. In practice, this means that we can only rely on the SCADA data and the data from the nacelle accelerometer. In the current contribution we focus primarily on the DEL estimation in the FA direction, as this is considered most relevant for the current jacket foundations. However, the methodology equally applies for SS direction.

## 2.2 Methodology

The main methodology of the present contribution can be understood as a *two-tier* neural network model. The first tier concerns itself with the generation and processing of relevant 10-minute features, which will serve as inputs for the second tier. In the first tier, an ANN model is utilized to estimate the thrust load on a 1s-basis. After this, the 1s thrust load, along with SCADA and accelerations from the SHM accelerometer, is processed into a variety of 10-minute metrics which, before being fed into the

second tier's ANN, undergo a dimensionality reduction procedure based on a feature selection algorithm. Then, in second tier, an ANN model is trained using the reduced set of input features, with the aim of estimating the tower FA DEL on a 10-minute basis. The motivation behind a 10-minute approach lies with the common framework for data processing (also 10-minute), the aim of including environmental effects and vibration levels, but also the issues inherent to working with different sampling frequencies ($1Hz$ SCADA, $12.5Hz$ accelerometer) and possible time-delays. We can observe the detailed methodology of the

current contribution in Figure 4.

### 2.2.1 Tier 1: 10-minute Feature generation

We can understand the first tier as globally contributing to generate and engineer relevant inputs (processed into 10 minute metrics) for the model employed in the second tier. A particular element of the current implementation of this first tier is the training, validation and employment of a thrust load estimation neural network model based on high frequent 1s SCADA. The

thrust load can be obtained from measurements by low-pass filtering the bending moment timeseries (with an upper frequency bound of 0.2 $Hz$).

Historical research carried on the OWI-Lab by Noppe et al. (Noppe et al., 2018a, b), has not only shown the relevance the quasi-static thrust load assumes in OWTs - albeit not being the sole contributor to fatigue - but also the possibility of replacing strain gauge use for the thrust load estimation and acceleration measurements (Cosack, 2010; Baudisch, 2012). The thrust load

is then not only an intrinsically relevant parameter in itself, as it is one of the major contributors for the structure's fatigue in modern OWTs (Noppe, 2019), but also because it synthesizes into a single parameter a vast amount of information pertaining to the SCADA data. Thus, the estimation and inclusion of the thrust load appears to be well-founded.

The estimation of the thrust load is, in this contribution, performed through the use of an artificial neural network. This ANN has been implemented using the tensor-manipulating framework TENSORFLOW, in particular its high-level API machine

learning library, *keras* (Chollet, 2017), implemented on the programming language PYTHON. The architecture selected for this ANN was a deep feed-forward neural network (Bishop, 2006). Research carried out by Schröder et al. (2018) into blade root flapwise damage equivalent load estimation has shown the greater performance of deep feed-forward ANNs when compared to

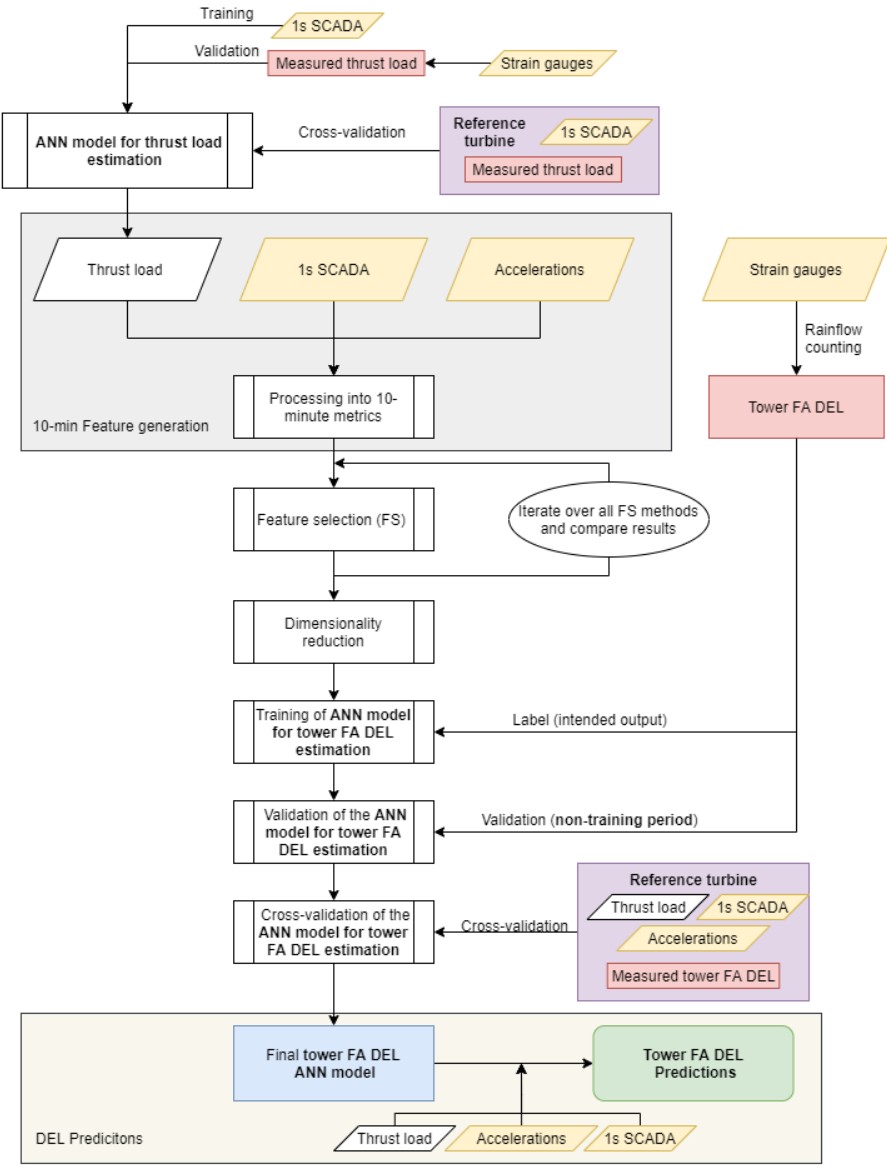

**Figure 4.** Flowchart representing the methodology's steps. Yellow highlights the sensors (and data by them captured), red the target, blue the final model and green the final results.



other surrogate models, such as polynomial chaos expansion (PCE). This relative better performance should also be translated into tower bending moment DEL estimation.

In this contribution, feed-forward neural networks (Hastie et al., 2009) are employed using rectified linear activation functions, commonly referred to as ReLU (Glorot et al., 2011; Jarrett et al., 2009), a standard non-linear activation function for regression-focused feed-forward ANNs. The loss (the model's error), was calculated by a mean squared error loss function, and is represented in Equation (2) and (3), for the DEL ($D$) and thrust load ($F$), respectively. Here, for a vector of $n$ predictions, we have $D_i$ or $F_i$, the vector of observed values, and $\hat{D}_i$ or $\hat{F}_i$, the vector of predicted values.

$$MSE = \frac{1}{n}\sum_{i=1}^{n}(D_i - \hat{D}_i)^2 \tag{2}$$

$$MSE = \frac{1}{n}\sum_{i=1}^{n}(F_i - \hat{F}_i)^2 \tag{3}$$

In order to minimize the loss function, an optimizer is required during the ANNs training. A common, well-performing choice is the adaptive moment estimation algorithm, also known as ADAM, or, in this particular case, ADAMAX, an extension of ADAM based on the infinity norm (Kingma and Ba, 2014). This particular optimizer was selected through hyperparameter tuning.

As mentioned, the ANNs' training will proceed until a given number of epochs is reached. A threshold of 100 was defined, as initial sensitivity analyses showed the model converging to a solution well before 100 epochs. This performance was monitored through the mean absolute error and root mean square error of the training and validation datasets (i.e. an independent dataset not used during training). In order to prevent the model overfitting for the training dataset, an early stop call-back mechanism with a patience of 10 epochs was implemented (Prechelt, 1998; Caruana et al., 2001).

The final thrust load estimation ANN topology had 4 hidden dense layers - i.e. 4 intermediary layers between the input and output layers where every neuron is connected to every other neuron in the next layer - with a varying number of neurons (from 64 to 300), achieved through hyperparameter tuning.

The thrust load, along with the SCADA and acceleration data, is then processed into 10-minute metrics. These include widely-known statistics as mean, minimum, maximum, median, mode (most common repeated value), standard deviation, range and root mean square (RMS) computed from ten-minute intervals of the timeseries. In addition also more atypical statistics such as spectral moments(1st – 4th (Miller and Rochwarger, 1970; Grimmett et al., 2020)), skewness and kurtosis are calculated from the timeseries. The inclusion of spectral moments, skewness and kurtosis in particular can be traced to Vera-Tudela and Kühn (2014). The formulae for these metrics can be found in Table A1, in the annex Section A. Apart from these *statistical* metrics, metrics that are rooted in fatigue assessment are also included. In particular, the damage equivalent moment (DEM) of the estimated thrust load is calculated, in a way representing the fatigue contribution of only the thrust load. In addition the damage equivalent acceleration is calculated by first cyclecounting the SHM accelerations - both the original signals and the signal 'transformed' into displacement. We can see them in Equation (4), where $a(t)$ stands for the



acceleration signal. The DEAs do not have any immediate physical meaning in terms of fatigue, but could be interpreted as a
*fatigue-weighed* mean amplitude of the acceleration.

$$DEA = \left( \frac{\sum_i n_i \cdot \Delta a(t)^m}{N_{eq}} \right)^{1/m} \tag{4}$$

The transformation of the original acceleration signals into displacements is performed through double integration in the frequency domain, as shown by Equation (5). Here, $\mathcal{L}$ stands for the Laplace Transformation, $a(t)$, the acceleration signal, $s$, the circular frequency $(2\pi \times f)$ and $x(t)$, the transformed displacement signal. This transformation is discussed in depth in
Maes et al. (2018).

$$x(t) = \mathcal{L}^{-1} \left\{ \frac{1}{s^2} \mathcal{L}\{a(t)\} \right\} \tag{5}$$

After the processing of all high-frequent signals (35) into 10-minute metrics, a total of 430 metrics is available for each 10-minute interval. Given the large number of variables some dimensionality reduction prior to the second tier of the algorithm is desirable.

### 2.2.2   Feature selection

Feature selection encompasses a number of methods focused in reducing the number of input variables of predictive models into the variables believed to be the most useful to the models (Leray and Gallinari, 1999). The reduction of input variables is frequently desirable as, by removing redundant variables, computational-, memory- and time costs are reduced (Guyon and Elisseeff, 2003). Moreover, it has been shown that dimensionality reduction may improve the overall performance of neural
network models, as non-informative variables can add uncertainty to the predictions and reduce the overall effectiveness of the model (Kuhn et al., 2013). ANN models' performance is thus highly dependant on the input data's quality, with certain inputs being vastly more relevant than others for the model's performance (Schröder et al., 2020).

Apart from enabling a dimensionality reduction, feature selection can also help identify the most important input-parameters. Knowing which features are relevant can be valuable information in assessing the added value of certain sensors, here in
particular the accelerometers, required signals from SCADA and the metrics that are worth calculating. As such the outcome of the feature selection may offer the possibility to optimize setups and reduce the amount of SCADA that has to be made available by the operator and processed.

Feature selection methods can be classified and grouped into various categories. Firstly, one can distinguish between supervised and unsupervised methods: if the outcome is not ignored (*i.e.* we have a target variable), then the technique is supervised.
This is precisely the case of the present contribution, wherein the target variable is the DEL of the tower FA bending moment, thus this contribution will solely focus on these methods. Supervised feature selection algorithms can be further sub-divided into Intrinsic, Wrapper and Filter methods.





Starting with the filter-based feature selection methods, these employ statistical techniques to evaluate the relationship between each input variable and the target variable, assigning a score for the relevance of the input variable. The scores obtained
for the relationship between the variable and the target are then used to choose (filter) the inputs that will be used in the model. Filter-based feature selection methods include Pearson's $r$, Dominance Analysis, Spearman's $\rho$, Kendall's $\tau$ and K-Best. These were computed using the PYTHON packages `scipy.stats`, `dominance_analysis` and `scikit-learn`.

Differently, wrapper feature selection methods generate several machine learning models evaluating different subsets of input variables, wherein the selected features are the ones that are present in the model that performs better, according to a
performance metric, such as the mean squared error (MSE), used in this paper. These methods' models use processes that add/remove predictors until an optimal combination that maximizes model performance is found. Unlike filter approaches, wrapper methods are able to detect the possible interactions between variables. There are, however, disadvantages, such as an increasing overfitting risk (for small samples) and a very significant computation time if the number of variables is large. Wrapper feature selection methods include Recursive Feature Elimination (RFE), in which a machine learning algorithm
present in the core of the model is fitted, the features ranked by importance, the least relevant features iteratively discarded and the model then re-fitted, with the processes being repeated until the required number of features is achieved. In this contribution, the machine learning algorithms used in the core of the RFE, also known as estimators, were a Random Forest (RF) regressor and a Decision Tree Classifier (DTC).

Finally, intrinsic methods represent feature selection models which possess built-in feature selection. This means that the
model only includes features that maximise the accuracy, thus performing automatically a feature selection during training. These include penalized regression models and decision trees, such as random forest algorithms which are an ensemble of decision trees algorithms that generate several decision trees during training and output the mean/average prediction of the individual trees. A deeper look into these methods can be seen in d N Santos et al. (2020b).

One final word should also be added regarding the popular methods Principal Component Analysis (PCA, Jolliffe and
Cadima (2016)) and Neighborhood Component Analysis (NCA, Goldberger et al. (2004)) and their non-inclusion in the present contribution. Although certainly powerful dimensionality reduction tools, both PCA and NCA are unable to provide insights into the original variables, as they transform the variable space (into principal components). It was deemed interesting to just work with the preexisting variables, as they are related to the installed sensors, and one can then learn more about their relative importance. This knowledge wouldn't be possible if PCA or NCA were to be employed.

### 270 2.2.3 Tier 2: Estimation of DEL using an artificial neural network

The second tier of the current contribution's model uses the features engineered in the first tier that were pick-up during the feature selection process as input. The measured tower bending moment DEL, attained from the strain gauges, is used as the label, or intended output, during the training stage.

Much like with the ANN used to estimate the thrust load, the tower bending moment DEL feed-forward ANN model is
implemented on PYTHON through the machine learning library *keras* - based on the tensor-manipulation framework TENSOR-FLOW. The activation functions between each neuron were, once again, rectified linear transfer functions and the loss function





MSE. The use of MSE as a loss function is particularly important in this application as MSE is more sensitive to outliers than other loss functions, which is relevant as the DEL is very sensitive to higher loads (Liano, 1996). For this particular application, it was found, through hyperparameter tuning and monitoring of the MSE, that the ADAM optimizer was the best performing
optimization algorithm. The final topology of the tower bending moment DEL ANN presented 6 dense hidden layers with 18-500 neurons.

After training, this model was validated on the same year as the training year (excluding the training period) and a different year, as well as cross-validated on the other fully-instrumented turbine for a whole year.

## 3 Results and Discussion

### 285 3.1 Thrust load model

Following the methodology prescribed in the previous section, one must first exhibit the results for the training, validating and cross-validating the artificial neural network that estimates the thrust load. The full results and discussion can be found in d N Santos et al. (2020a). The high-frequency SCADA data used to train consisted in wind speed ($m/s$), rotor speed ($cps$), mean pitch (°), nacelle orientation (°) and actual active power ($kW$) from 12 days, carefully selected as to be representative of all
operating conditions. Strain measurements from the same time period undergo a temperature compensation before calculating the resulting bending moments and filtering this last signal with a low pass filter with an upper frequency bound of $0.2\ Hz$. The filtered FA bending moment $M_{tn}$ signal is subsequently translated into thrust load, which must further be corrected for the air density.

The model training, monitored through the Mean Absolute Error (MAE) and Root Mean Square Error (RMSE) was deemed
satisfactory, as convergence was achieved with both MAE and RMSE below 1 %. We can observe the model's output when plotting a discrete time series of interest, e.g. when the turbine is operating at rated power (see Figure 5), and we compare the predictions with the measured values of the thrust load. In this figure, we can observe how close the predictions accompany the measured thrust load, capturing almost fully the base quasi-static loading behaviour.

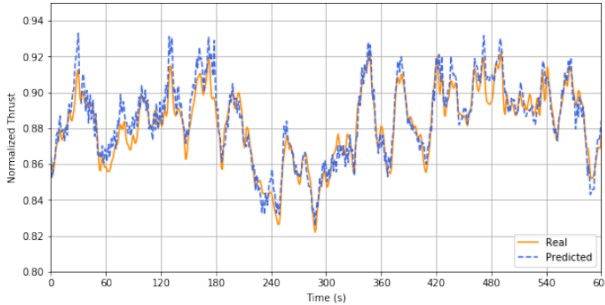

**Figure 5.** Measured (orange) and predicted (blue) thrust values for a 10 minute time instance at rated power.





The model was then validated for 3 months of data outside of the training period on the same turbine as for training. The

model was also applied to a different (non-training) wind turbine with a similar SHM setup, the cross-validation stage. The results for predicted thrust load were compared with the measured values attained from the strain gauges at this location. This cross-validation served to assess whether the model is transferable to other turbines in the farm. In Figure 6 we can observe the model's performance (through the mean absolute error expressed as a percentage of the maximal thrust load value) for both the validation and cross-validation datasets plotted against the wind speed (MAE binned according to the wind speed with a step

of 2 $m/s$).

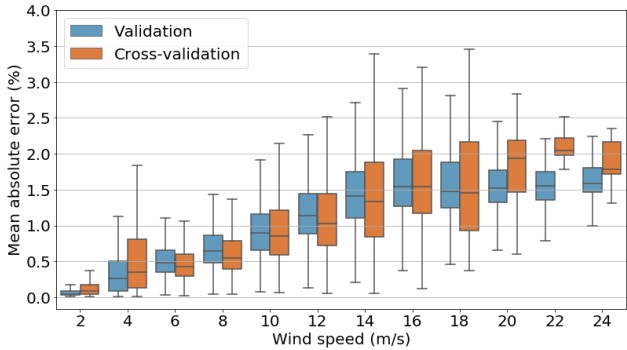

**Figure 6.** Box plot of MAE of thrust load estimations, expressed as a percentage of the maximal training thrust load value (binned) for the validation and cross-validation turbines plotted against wind speed.

For both cases (validation and cross-validation) we can observe that both curves are rather similar, with one and the other presenting a slight increase in the MAE as the wind speed increases and most values staying below 2 %. The increase of MAE with the wind speed is expected, as higher loads are attained for higher wind speeds (and the MAE relates to the absolute value of the loads). The two curves differ in the fact that, for the cross-validation, the MAE box plots present slightly larger whiskers.

It was also identified, for the cross-validation, a small number (3) of overshoots prior to a rotor-stop, not present in the training dataset, and whose cause remains undetermined. We can understand the higher maximal value for the cross-validation (attained for 4 $m/s$) as being related to these overshoots prior to a rotor stop.

Regardless of the presence of these overshoots, they do not overly influence the final results, as both the original and cross-validation turbines' MAE box plots, remain highly similar. This similitude allows us to affirm the transferable nature of the

ANN model to other offshore wind turbines.

### 3.2 Parameter and sensor significance

A key concern of the current contribution is to understand the added value each sensor brings to the predictive model, in particular, the contribution of the dedicated tower SHM accelerometer. Apart from enabling to reduce the input variable space's dimensionality, the performance of a feature selection routine also allows us to identify the most important parameters engi-





neered in the first tier (*cf.* Section 2.2.1 and 2.2.2) and linking them with the original sensor. Which in turn allows to assess whether a particular sensor is worth adding to the setup.

Finally, in order to assess the gains the current instrumentation layout presents relative to less-instrumented turbine setups, a comparative analysis based on ANNs is carried out for 8 different instrumentation scenarios.

### 3.2.1 Feature selection

As mentioned above, in order to understand which engineered features are relevant to determine the tower bending moment DEL (and thus, which sensors are important), several feature selection algorithms are studied and the groups of features they select registered. Before performing any feature selection routine, one must first engineer the necessary 10-minute features.

The thrust load, accurately estimated as detailed in Section 3.1, can then be, along with the acceleration and SCADA signals (in a total of 35 high-frequent parameters), processed into 10-minute metrics, as seen in Section 2.2.1. The different
permutations between signals and metrics generate a total of 430 features available at ten minute intervals.

This large number of features further elicits the need to perform a feature selection study, in order to lead to reduce the number of the input variables. Several methods were compared, including filter-based methods (Pearson's $r$, Dominance Analysis, Spearman's $\rho$, Kendall's $\tau$, K-Best), wrapper-based methods (Recursive Feature Elimination (RFE) with either a decision tree classifier or a random forest estimator - feature ranking algorithm) and an intrinsic method (random forest), described in
Section 2.2.2 . In Table 1 we can observe each feature selection method, along with the number of features (also expressed as a percentage of total amount of features) selected by each method and the class. The full results of the feature selection are shown in Annex B in Table B2. Table B1 provides a quick explanation of the nomenclature employed. A more in-depth discussion can be found in d N Santos et al. (2020b).

**Table 1.** Feature selection methods, classes and number of features selected.

| Method | Number of features selected | Method class |
|---|---|---|
| Dominance Analysis | 50 (11.63%) | Filter-based |
| Pearson's $r$ | 134 (31.16%) | |
| K-Best | 145 (33.72%) | |
| Spearman's | 191 (44.42%) | |
| Kendall's | 196 (45.58%) | |
| RFE RF | 20 (4.65%) | Wrapper-based |
| RFE DTC | 22 (5.12%) | |
| Random Forest | 18 (4.19%) | Intrinsic |

Table 1 shows a clear distinction between filter-based and other methods, wherein the former selects 100+ features. This
might be because filter-based methods aren't non-linear, which might increase the difficulty in discerning the most important




features and effectively withholding non-relevant ones. Nevertheless, to properly assess the performance of each feature se-
lection method, the features selected by each feature selection method were fed into a generic feed-forward artificial neural
network. The ANN presents two hidden layers with 50 and 100 neurons, respectively, using a rectified linear activation func-
tion between each layer. The number of neurons in the input layer is equal to the number of features selected by the feature
selection method. The ANN also applies an Adamax optimizer and a mean squared error loss function with the target being
the tower FA bending moment DEL, as well as a limit of 100 epochs. The training-testing split was performed through k-fold
cross-validation with 10 folds and a batch size of 5, which allows us to assess the generalization of each model (Anguita et al.,
2012). The results for each method are presented in Figure 7. The spread in results for each method is due to the 10-fold
cross-validation and methods' classes are distinguishable through a colour scheme: green (intrinsic), orange (wrapper) and
blue (filter).

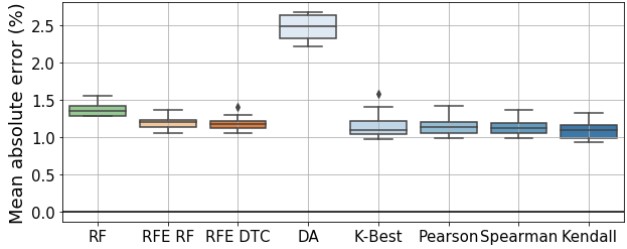

**Figure 7.** Mean absolute error of every feature selection method's ANNs as a percentage of the maximal DEL value. RFE stands for recursive
feature elimination, RF for random forest, DTC for decision tree classifier and DA for dominance analysis.

In Figure 7, we can observe that all methods present similar values (between $1.5\%$ and $1\%$), apart from Dominance Analysis.
Upon first glance, one would be tempted to assume that Kendall's $\tau$ is the best model, as it presents a marginally lower mean
MAE than other methods. However, it is also crucial to look at the number of features selected by each model (*cf.* Table 1). The
intrinsic and wrapper methods selected around 20 features, whereas filter-based methods (apart from Dominance Analysis,
which selects 50 features) all selected above 100 features. For the latter, the higher the number of features, the better the
model's performance. This evidenced by Figure 8.

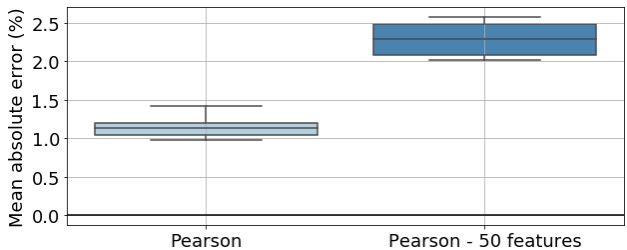

**Figure 8.** Comparison of mean absolute error between model trained with features selected by Pearson's $r$ and just the first 50 features
selected.





In Figure 8 we can see that, for Pearson's $r$, if, instead of 134 (31,16% of the total amount of features) features selected, there are only 50 (11,63%) features selected, the MAE increases by over 1 % (attaining a similar value to Dominance Analysis).

Given the results shown in Figure 7, we are then in the presence of a trade-off scenario: either we have a small number of highly relevant features selected by recursive feature elimination, and thus, a high computational cost in feature selection, but smaller cost in neural network model training and input variables processing, or we have a high number of features selected by filter-based methods, which entails, a small computational cost for feature selection and a bigger cost for neural network model training.

Naturally, as the objectives of Section 3.2 are to reduce the dimensionality of the input variable space and gain insights into the most important features (and therefore, the most important sensors), filter-based feature selection methods ought to be discarded in favour of intrinsic or wrapper methods. These latter, for a low amount of features (around 20) present models which perform as well as models involving higher numbers of features.

If we now focus on wrapper methods (*cf.* Table B2), we notice that these select solely FA and X accelerations/displacements signals, whilst also selecting RPM, Thrust and wind direction. These methods pickup metrics such as kurtosis and skewness, but also DEA, DEM, RMS, range, std, max, min, mean and the $2^{nd}$ and $4^{th}$ spectral moments. As for intrinsic methods, they only selected metrics for bottom-, mid- and upper-level FA displacements (DEA, max, min, range, $2^{nd}$ and $4^{th}$ spectral moments) and for thrust (DEM, max, std, $2^{nd}$ and $4^{th}$ spectral moments).

Globally, we can say that some metrics are selected more often by all methods - DEA, RMS, range, min, max and std - and that some features are selected by all methods - DEA of FA displacement for bottom-, mid- and upper-level, range of FA bottom-level displacement, min of FA upper-level displacement and DEM of Thrust.

The fact that all methods select DEA FA displacement metrics and the DEM of Thrust is understandable, as this was the initial assumption behind the cyclecounting of these signals - that they would more closely relate with the target, the tower FA bending moment cycle counted DEL. Likewise, the more often selected statistics (RMS, range, std, min, max) relate to variations within the signal, which is the most relevant in the cycle counting process.

When focusing on SCADA-related features, we can observe that Wrapper (RFE) and Intrinsic methods select SCADA-related metrics which weren't picked-up by filter methods - mean, max, range, std, $2^{nd}$ and $4^{th}$ spectral moments of Thrust, but also std and rms of RPM and min wind direction. The higher number of thrust-related metrics selected might be due to its initially linear relationship to the tower FA bending moment DEM (see Figure 9 where, especially for lower wind speeds there's a linear correlation). SCADA parameters as the wind direction are only picked-up by the wrapper methods. These, along with the intrinsic method, also signal more the spectral moments. Again, this is to be excepted, as filter-based methods, even 'nonlinear' ones such as Spearman (dependant on monotonic functions) and Kendall, can't pick-up the more complex interactions between the SCADA data and the target variable. Only two parameters weren't chosen by any feature selection algorithm - the temperature and pitch. Nevertheless, if we discard filter-based methods, SS and Y SHM accelerations appear to be unnecessary, along with the low-quality nacelle-installed accelerations, power and wind speed. Likewise, only 3 metrics weren't select by any method - median, mode and the $3^{rd}$ spectral moment - in future uses of this methodology, these needn't be considered.

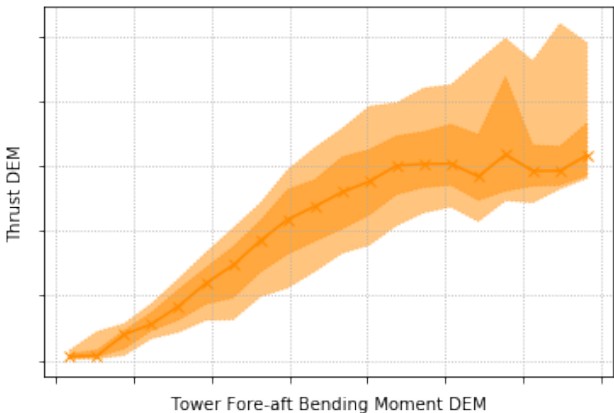

**Figure 9.** Thrust DEM *vs.* tower fore-aft bending moment DEM.

In sum, the more trustworthy wrapper and intrinsic methods give us then more meaningful features; this has lead to the selection of RFE (DTC) for further use (in Section 3.2.2, 3.3), instead of RF or RFE RF, as it presented a better, albeit marginally so, performance. We can also clearly affirm that one ought to engineer a varied set of features (including spectral moments, kurtosis, skewness, RMS and cycle counting the signal) stemming from the SHM accelerometers (for FA and X-direction signals, which should also be translated into displacement) and the thrust load estimation (with a desirable inclusion of certain SCADA-related features, such as wind direction and RPM). It ought to be reinforced that the conversion of SHM accelerations to displacements appears to be crucial. Metrics as median, mode and the $3^{rd}$ spectral moment seem to bear no fruits, as do the Y and SS SHM accelerations, low-quality nacelle-installed accelerations, temperature and pitch. The SHM acceleration parameter with most metrics selected was the FA bottom-level displacement signal, which might mean that this is the most important placement for the accelerations. A surprising result as it was expected that the top side accelerometer would be a better placement. Nevertheless, in order to identify the most important placement of the SHM accelerometer (if at bottom-, mid- or upper-level), it would be desirable to perform a dedicated study with 3 scenarios - with only one SHM accelerometer at a level per scenario. This has to, for the time being, be left for future research tasks (especially for monopile foundations).

In this section, several feature selection methods were compared, with a clear preference for wrapper or intrinsic methods, and the relevant metrics and sensors (and signals) identified, with it being then apparent that, when faced with an instrumentation-scenario as described in Section 2.1, nacelle-installed low-quality accelerometers should be disregarded in favour of SHM accelerometers, the calculation of FA and (to a lesser extent) X displacement metrics prioritized over SS and Y and that the estimation of thrust load is paramount.

### 3.2.2 Minimal instrumentation study

In the previous section we have seen how the most valuable sensors are the dedicated tower SHM accelerometers, along with the estimation of the thrust load based on 1s SCADA, and which metrics are more relevant. However, in order to fully





answer what is gained with the addition of each sensor and by their increased quality, an additional analysis is required. This
is particularly relevant, as the instrumentation setup described in Section 2.1 isn't always present, or even the most commonly
present in wind farms.

To this point, eight different plausible scenarios are defined. Each scenario only considers a subset of all parameters to
be available, in line with scenarios OWI-lab has experienced in past projects (scenarios involving 10-minute SCADA and
low-quality nacelle accelerations are typical for older wind turbines, whereas the inclusion of 1-second SCADA and/or SHM
accelerometers is sometimes seen in newer turbines). These scenarios are identified in Table 2. For each scenario a model is
trained and is tested for a validation and a cross-validation dataset, both comprising in total of one year worth of data. The
validation dataset comprises of the dataset of the year of the ANNs training and the cross-validation comprises of a dataset
of one year of another, non-training turbine. Both results are presented together, in which the cross-validation results allow to
assess to what level the results can be transferred from one turbine to another.

Scenario H, the final model, follows the methodology prescribed in Figure 4 and, as presented in Section 3.2.1, uses the data
selected using the RFE DTC feature selection algorithm (the only scenario where feature selection was performed), including
components from 1s SCADA, dedicated tower SHM accelerometers and the thrust load. The RFE DTC was the chosen method,
as explained in the previous section. Scenario F - 1s SCADA, thrust load an low-quality nacelle accelerations - is the prevailing
instrumentation scenario throughout the farm in this contribution, except the aforementioned two fully instrumented turbines
(scenario H), and will be further discussed in Section 3.4.

**Table 2.** Scenarios investigated. Different scenarios imply that different data sources are considered to be available, in line with real world
experience. E.g. Scenario A considers that only 10 minute statistics of SCADA are available while scenario H covers a near ideal situation
only possible on the two SHM turbines. Scenario F is possible at all turbines of the particular farm in this contribution. Note that the access
to 1s SCADA inherently implies access to 10-minute SCADA. Some scenarios exclude the calculation of the 1s thrust load estimation to
assess the added value of calculating this type of parameter.

| Model | Scenario | | | | |
|---|---|---|---|---|---|
| | 10-minute SCADA | 1-second SCADA | Thrust estimation (1s SCADA) | Nacelle low-quality accelerometer | SHM accelerometer |
| A | X | | | | |
| B | X | | | X | |
| C | | X | | | |
| D | | X | | X | |
| E | | X | X | | |
| F | | X | X | X | |
| G | X | | | | X |
| H (final model) | | X | X | | X |





In Figure 10, we can see the absolute error boxplots, expressed as a percentage of the maximal training dataset DEL, with the MAE also indicated. Here, the absolute error relates to the absolute values, and not the modulus, and is given by the difference between real and predicted values, $X_i - \hat{X}_i$ (presented as a percentage of the maximal DEL).

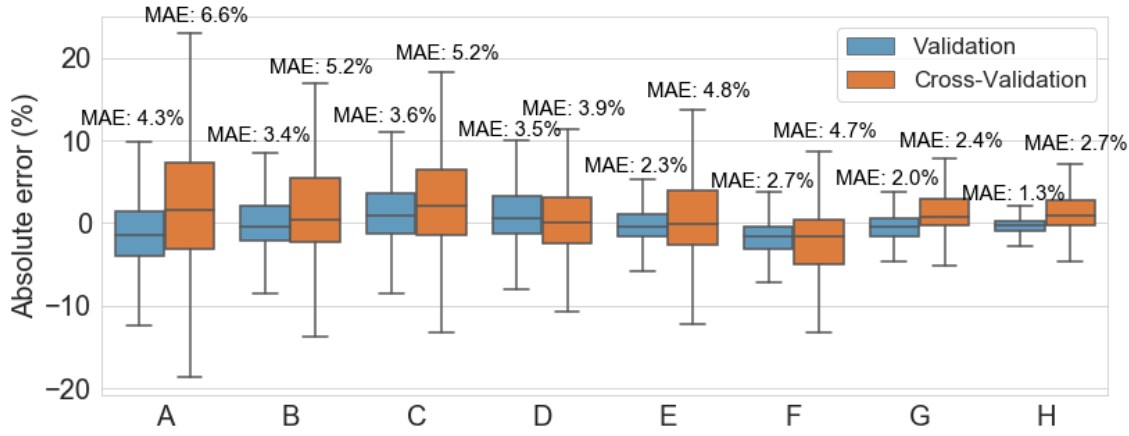

**Figure 10.** Comparison of the ANN model's performance in validation and cross-validation for eight different sensor data quality scenarios.

Some general remarks can be drawn by analysing this figure. Firstly, when comparing the first four scenarios (A-D), we can see that the inclusion of 1s SCADA induces a better model, with a lower MAE (specifically if we consider the cross-validation).
Both for 10-min SCADA and for 1s SCADA the inclusion of nacelle-installed low-quality accelerometers (B and D) doesn't necessarily enhance the performance of the model itself (validation MAE, in blue, doesn't improve that much, especially for 1s SCADA), but it does improve the model's generalization to other turbines (cross-validation MAE, in orange, almost decreasing by 2 % for both cases). We can try to understand this as implying that, albeit for a single turbine SCADA data alone might provide a decent enough model, if we want to apply a model to different turbines, then the inclusion of accelerations is
paramount.

Focusing on 1s SCADA, we can see that the inclusion of the thrust load (E and F) further improved the model itself (validation), but isn't able to generalize as well for the cross-validation turbine, actually slightly degrading the results when compared to 1s SCADA and nacelle-installed accelerations (D). Here, we might understand that the thrust load is further introducing some site-dependency related to wind and controller conditions, thus overfitting for the training turbine. Pivoting back to 10-
min SCADA, if we include dedicated tower SHM accelerometers (G), again we see accelerations improving the generalization, but this time, also vastly improving the models performance (MAE close to 2 %, error well between the minus and plus 10 % bounds). This might be due to the fact that, with more accurate accelerometers, with greater quality, a fuller picture of the structural dynamics can be drawn for both the validation (training) and cross-validation turbines. Scenario G - 10-minute SCADA and SHM accelerations - should also be highlighted, as it is performing rather well (both MAEs around 2%). This is
especially relevant for older wind farms, where usually 1s SCADA isn't available. Finally, the results for the model based on the RFE DTC selected features are clearly the best (H). This final model includes the thrust load which, as discussed, while



improving the model's performance, might degrade generalization (adaptability to other turbines). As a 1s SCADA with SHM accelerations model that does not include the thrust load is out of the scope of the current methodology, it was deemed relevant to further study this phenomenon *a posteriori*. It should, however, be said that, in itself, the thrust load is a highly relevant

parameter, be it standalone, or if intended for further use in training an ANN model.

We can then say that the best models include data from SHM accelerometers and are able to present MAEs of around 2 %. 1s SCADA models can have their performance enhanced by estimating the thrust load or including nacelle accelerations and have MAEs of around 4 % for cross-validation, with most errors falling within ±10 %. For just 1s SCADA or 10-minute SCADA models (A,B,C), MAEs are above 5 % and most cases will fall within ±20 %. It is up to the operator, then, to determine which

bounds are considered acceptable.

The present contribution provides a detailed overview of how the ANN-based methodology might perform for different instrumentation scenarios on jacket-foundation offshore wind turbines. It is expected that, for monopile-foundation OWTs, where the importance of wave-related dynamics is much greater, other conclusions will be drawn as to the minimal instrumentation setup performance.

### 3.3    Fatigue rate (DEL) estimation

Following the discussion held in the preceding section( Section 3.2), a deeper look into the best performing model - scenario H - is required.

For this, several ANN topologies were tested employing the Adam optimizer and a mean squared error loss function, a limit of 100 epochs, k-fold cross-validation with 10 folds, a batch size of 5 and monitoring of MSE and RMSE. Of the different

tested topologies, the best performing one presented 5 hidden layers with 100, 200, 300, 200 and 100 neurons, respectively. The data selected for training consisted in a randomized sample of 8000 datapoints within a given year (the training dataset represented 35,7 % of the available data for that one year, with a 80%-20% train-test split). The deep neural network was then trained on this dataset and its performance tested on the testing dataset, and then validated on a different period in the same year outside the training period, a different year and cross-validated for an entirely different turbine. In Figure 11 we can see

plots of the timeseries for normalized DEL (both for the validation and cross-validation datasets) for a period of 7 days. We can see how, for both cases, the predictions closely accompany the measured behaviour.

In Figure 12, the absolute error of the tower FA bending moment DEL predictions, estimated as for Figure 10, expressed as a percentage of maximal value of DEL (for training) has been plotted for the training year, a different year (but still in the

training turbine) and a different turbine. Additionally, the MAE for each case (also expressed as a percentage of the maximal training DEL), has been superimposed to the plot.

Here we can see that the model performs rather well for the training year – the MAE is kept around 1 %. Likewise, for a different year, the model performs rather well, with a MAE of 1,7 %, although a slight underprediction is noticeable (a positive error means that the real value is superior to the prediction, also existent for the training year) and there is a marginally

bigger spread. Nevertheless, differences for the performance between the training year and different year are almost negligible.



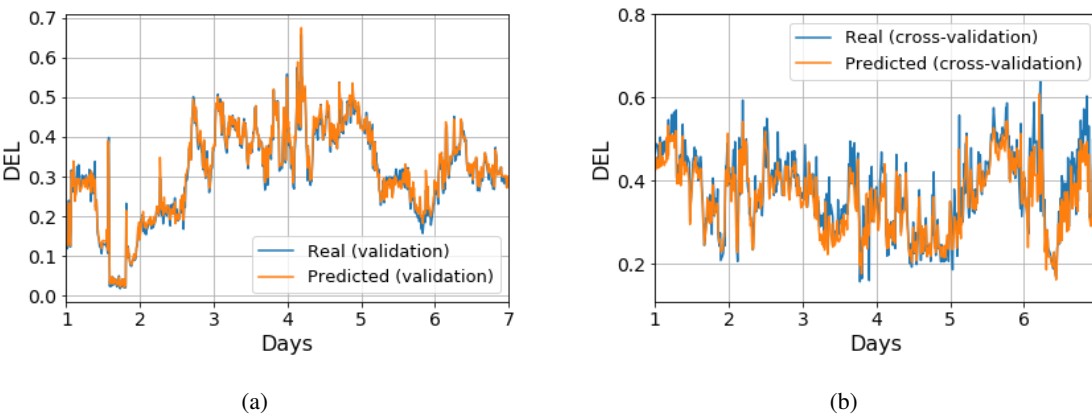

**Figure 11.** (a) Timeseries of DEL for 7 days of the validation dataset (Real - blue; Predicted - orange). (b) as in (a), but for the cross-validation dataset (different turbine from training).

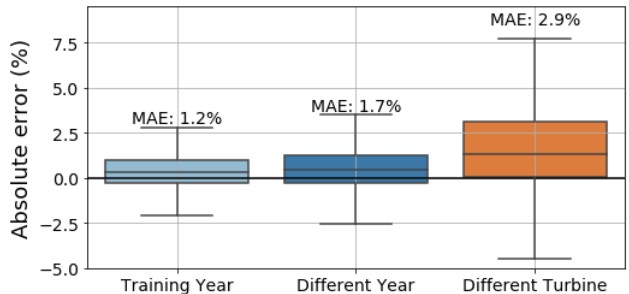

**Figure 12.** DEL prediction error for validation (training year and different year) and cross-validation, expressed as a percentage of the maximal DEL.

The model then seems to properly adapt and generalize. We can thus conclude that the model is performing appropriately, presenting sufficient generalization to successfully predict for datasets which do not correspond to its training period, with a MAE around 1%, which is within the bounds defined established Section 3.2.1.

Things however change for a different turbine: the MAE increases about two-fold to 2.9 %, the spread for the error is much more noticeable, and there is an accentuated underprediction (positive error). There may exist several, non-exclusive, reasons behind this degradation of the model's performance for a different turbine than that of training. Firstly, one can not ignore the inherent site-dependency of the trained model: the ANN has been trained on a given turbine, so it is expected that, even though it's able to capture the overall behaviour of other turbines of the same farm, it will have a greater difficulty to adapt to different turbines as well as for the training turbine. Additionally, as seen in subsection 3.2, the inclusion of thrust-related metrics might
further enhance site-dependency. Furthermore, this site-dependency can be increased by the feature selection process - the features that are selected are for the training turbine; these might possibly differ for the cross-validation turbine. One possible way to circumvent this would be to have a population-based model (Antoniadou et al., 2015; Worden et al., 2020) which would



use the data of both turbines during training. However for our current study, this would impede the cross-validation as only two turbines are available. Nevertheless, we can still affirm that, albeit the relatively worse results for cross-validation (different
turbine), the model still performs within the realms of acceptability, giving us a certain degree of trustworthiness.

In Figure 13, we can take a deeper look at the model's performance, wherein the absolute error of the tower FA bending moment DEL predictions, is binned according to the wind speed with a step of 2 $m/s$ for the validation dataset (year of training) and the cross-validation dataset (different turbine).

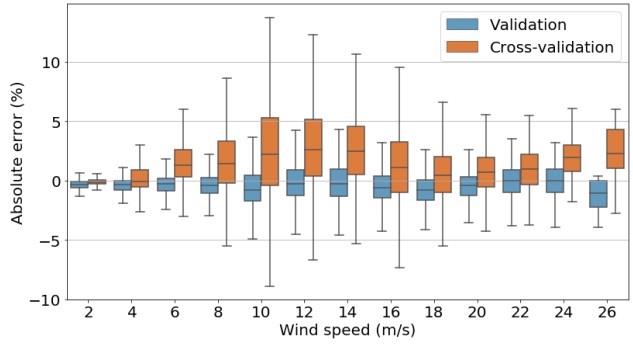

**Figure 13.** Box plot of absolute error (binned) of DEL for the validation and cross-validation turbines plotted against wind speed.

Here, we can verify the better performance of the model for the validation dataset, as evidenced by Figure 12 and that
the model is under-predicting for the different turbine (cross-validation). More interestingly, we can observe how the model performs worse for a different turbine for mid-range speeds (8-18 $m/s$) rather than for higher speeds. This is is possibly related with the worse performance of the model under wake. In Figure 14 we can observe how even for the training turbine (OWT 7, *cf.* Section 3.4), a similarly higher error appears for the training turbine when operating under wake. Even though the amount of data is noticeably lower (thus, there are only boxplots up to 18 $m/s$), we can also see the higher errors above 10 $m/s$.

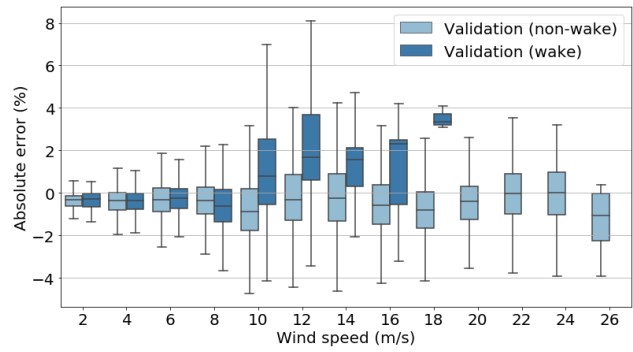

**Figure 14.** Box plot of absolute error (binned) of DEL for the validation/training turbines plotted against wind speed under wake and under free-flow conditions.



This worse performance for the cross-validation, might be possibly due to a lack of under-wake data, could be enhanced through wake transfer-models or a population-based approach. Nevertheless, we can say that, overall, the tower bending moment FA DEL ANN model is performing well and presents itself as a viable solution for DEL estimation under a fleet-leader concept.

## 3.4   Farm-wide

As mentioned in Section 3.2, most turbines present in the wind farm dealt with in this contribution do not possess fully-instrumented setups (scenario H, *cf.* Table 2), but rather 1s SCADA (which allows to obtain the thrust load) and low-quality nacelle-installed accelerometers - scenario F (this data must be processed into 10 minute metrics, as described in the methodology). Therefore, in order to perform a farm-wide DEL assessment based on a fleet-leader concept (Noppe et al., 2020), the tower bending moment FA DEL ANN model is employed for the instrumentation specs of scenario F. Figure 15 schematizes
the pipeline for the farm-wide model.

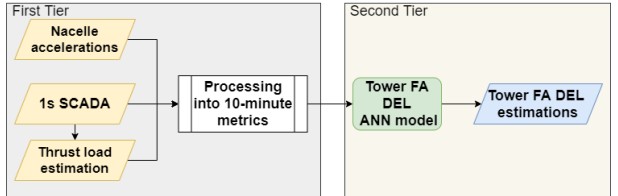

**Figure 15.** Schematic of tower FA DEL estimation ANN methodology.

   The ANN model methodology presented in the section above can be put to use in a real-world farm-wide setting with 48 assets, given that there's sufficient data from the sensors - in this case, 1s SCADA (including nacelle-installed accelerations and estimating the thrust load). It ought to be signalled that the ANN model was trained in OWT 7 and cross-validated in OWT 31. We can then estimate the DEL for each turbine for 4 months in the Summer of 2020, and plot farm-wide based on 10-minute
averages, as seen in Figure 16:

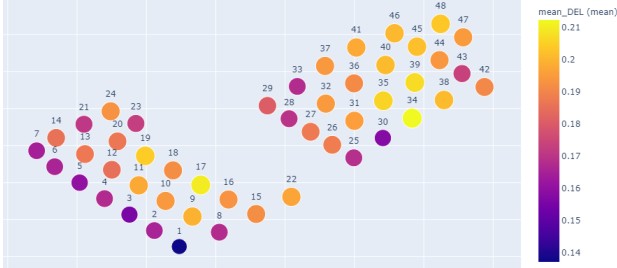

**Figure 16.** Mean DEL farm-wide plot, normalized in relation to the highest training DEL value.





Here, we can observe that the DEL increases from east to west, which is in accordance with the dominant south-west (SW) wind direction in the Belgian North Sea. This is related to the effects wake has on DEL, as turbines under free-flow conditions present lower DELs than their counterparts under wake.

There are however some noticeable exceptions. Several turbines have a lower than expected DEL (in blue-purple in Figure 530 16). Although there are many underlying phenomena that might contribute to this behaviour, we can start by analysis the power production farm-wide for the same period (Figure 17):

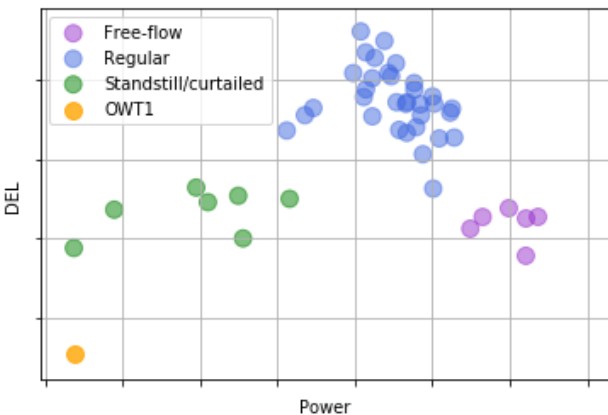

**Figure 17.** DEL *vs* power plotted for the mean values of every turbine of the farm. The lowest DEL turbine - OWT1 - is signalled.

We can see here that there's generally a positive correlation between the turbines which face a lower DEL and the ones that produce less. If we focus on OWT 1 (in orange), which presented the lowest DEL and is on the lower end of the power-axis in Figure 17 - we observe that the reason it doesn't produce so much is due to it being in standstill for a sizeable amount of time 535 during the period investigated. This more frequent standstill, along with OWT1 usually facing free-flowing wind due to the dominant wind direction, produces the lowest mean DEL of the farm for the period under study. Figure 17 further highlights two distinct sets which present a lower mean DEL. Firstly, turbines that were in standstill/curtailed, in green, of which OWT30 is an example, wherein the lower power output (and thus, lower amount of time functioning) is correlated with lower DELs. Secondly, turbines that were located in the first string (row of turbines) facing the dominant wind and thus, faced mostly 540 free-flow conditions, which enabled both a higher production (higher mean power) and lower DELs, as wake-induced load variations weren't present.

Apart from the turbines facing a lower DEL, there are two other notable exceptions with a higher than expected DEL: OWT 17 and 34 (the two turbines on top of the blue curve of Figure 17). We can try to understand the underlying reason why these turbines are popping-up through some farm-wide plots. If we begin with Figure 18, where the fore-aft nacelle accelerations 545 standard deviation is plotted farm-wide, we can see that both OWT 17 and 34 are the turbines that present the highest standard deviation for the FA nacelle accelerations. This makes sense, as more variability in the measured accelerations is related to higher DELs (larger cycles).




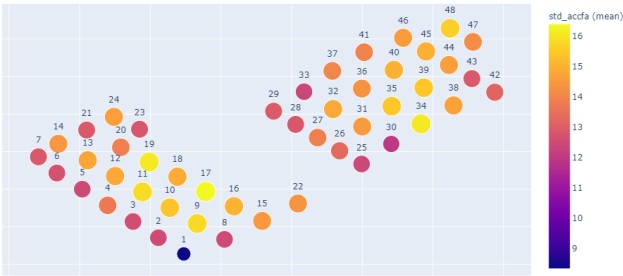

**Figure 18.** Farm-wide plot of fore-aft nacelle accelerations standard deviation.

Further focusing on OWT 17, Figure 19 shows us clearly that, for the farm-wide plot of the mean RPM range, this turbine is clearly highlighted. This might mean that the higher DEL faced by OWT 17 is related to a higher mean range for the RPM, which can then mean that the variations on the rotational speed of the rotor are higher, thus increasing the global DEL. This is also apparent in a higher DEM of the thrust load for OWT 17.

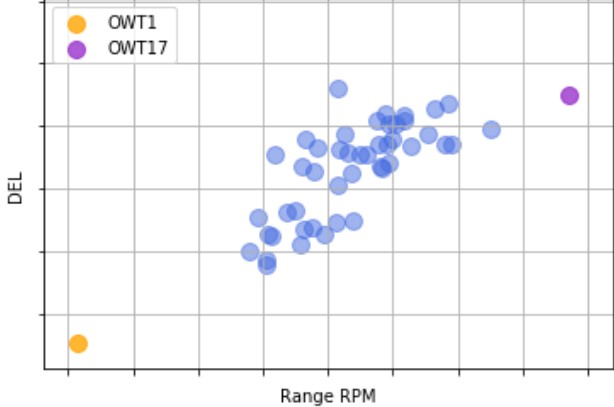

**Figure 19.** DEL *vs* range of RPM plotted for the mean values of every turbine of the farm. The lowest DEL turbine - OWT1 - is signalled, as is OWT 17.

Regarding OWT 34, no apparent unique answer for the higher DEL could be unveiled in the SCADA- or acceleration-based features. However, as seen in Figure 18, the accelerations appear to highlight a bit more clearly OWT 34. The answer might indeed lie there - the underlying causal phenomenon might not be picked-up by SCADA, but only by the accelerations. It is interesting to note that this turbine in particular presented an unbalanced rotor for this period. The link between acceleration measurements and a higher DEL might lie there. Nonetheless, the 10-minute ANN tower FA DEL model is composed by numerous input variables, wherein discerning a single culprit might be difficult, as some behaviours might be the fruit of complex interactions between several parameters.





## 4   Conclusions

In the current contribution a methodology to determine jacket-foundation OWTs tower bending moment DEL based on artificial neural networks has been successfully implemented. This value can serve wind-farm operators in taking informed lifetime related decisions. The model has been validated on two distinct years of data for the training turbine, and presented a MAE of around 1 %. Its cross-validation on a different turbine, albeit performing slightly worse (MAE close to 3 %), has proven the applicability of this model to other turbines.

The model uses a reduced set of input data determined by the employment of the recursive feature elimination (using a decision tree classifier estimator) feature selection algorithm. This data is provided by 1s SCADA (along with the estimation of the thrust load) and SHM accelerometers sensors. A comparative study of feature selection techniques and another of different sensor setups, based on the use of neural networks, have allowed to identify the sensors and engineered features of greatest importance, and which sorts of error might be related to each instrumentation setup. From this study, it appears that

the installation of dedicated tower SHM accelerometers is advisable, as, if a fleet-leader model is to be applied, acceleration data is essential to a good farm-wide generalization.

Finally, this methodology is employed on a farm-wide setting, allowing to identify turbines with outlier DEL values. By looking at the SCADA and acceleration input data, tentative answers for outlier behaviour can be drawn, providing farm operators with important insights related with maintenance.

## 5   Future Work

Throughout this contribution, the development of this methodology has seen some questions rise, which deserve a deeper look. These questions can be see as future steps in research. Some future research directions include:

– Application of the current methodology to monopiles;

– Employment of other machine learning algorithms (e.g. SVM, decision trees, kringing, gaussian process regression) and
comparative performance study;

– Study what's the contribution (or lack thereof) of the inclusion of the thrust load in the final DEL model;

– Understand the impact of sensor quality (10-min, 1s, etc.) in a monopile-foundation OWT;

– Develop a population-based strategy: training not dependant on only one turbine, but on data from possibly several turbines;

– Utilize the DEL ANN model to research specific phenomena, such as wind-wave misalignment and wake effect on monopile-foundation OWTs;

– Evaluate which SHM accelerometer placement (bottom-, mid- or upper-level) is the most valuable.





*Code availability.* The code has not been made publicly available.

*Data availability.* As the data proprietary is an industrial partner of this project, the data used in this paper cannot be made publicly available

**Appendix A**

**Table A1.** Statistical metrics formulae applied to parameter $x$ in which $N$ is the number of individual samples $x_i$ in a ten minute interval.

| Metric | Formula/Definition |
|---|---|
| Minimum | Smallest value of the set |
| Maximum | Biggest value of the set |
| Mode | Most frequently observed value of the set |
| Mean ($\mu$) | $\sum \frac{x_i}{N}$ |
| Median | $(\frac{N+1}{2})^{th}$ number of the set |
| Standard deviation ($\sigma$) | $\sqrt{\frac{1}{N}\sum_{i=1}^{N}(x_i-\mu)^2}$ |
| Range | Difference between the lowest and highest value |
| Root mean square (RMS) | $\sqrt{\frac{\sum_{i=1}^{N}x_i^2}{N}}$ |
| Spectral moment $i$ ($m_i$) | $\frac{1}{N}\sum_{k=1}^{N}(x_i-\mu)^i$ |
| Skewness ($g_1$) | $m_3/m_2^{3/2}$ |
| Kurtosis ($g_2$) | $m_4/\sigma^4-3 = m_4/m_2^2-3$ |

**Appendix B**





**Table B1.** Variable abbreviation explanation.

| Abbreviation/Symbol | Meaning |
| --- | --- |
| ACC | Acceleration Sensor |
| BL | Bottom Level |
| ML | Mid Level |
| UL | Upper Level |
| FA | Fore-Aft |
| SS | Side-to-Side |
| X | X-Direction |
| Y | Y-Direction |
| disp | Displacement |
| dea | Damage Equivalent Acceleration |
| dem | Damage Equivalent Moment (for Thrust) |
| $g_2$ | Kurtosis |
| $g_1$ | Skewness |
| $m_i$ | $i$-th spectral moment |
| std | Standard Deviation |
| Temp | Temperature |
| ∘ | Pearson's correlation coefficient, $r$ |
| △ | Spearman's correlation coefficient, $\rho$ |
| □ | Kendall's correlation coefficient, $\tau$ |
| ∗ | Dominance Analysis |
| † | K-Best Selector (F-regression) |
| ⋆ | Recursive Feature Elimination (cross-validation) using a Random Forest Estimator |
| ♦ | Recursive Feature Elimination (cross-validation) using a Decision Tree Classifier |
| × | Random Forest Estimator |



Table B2: Comparative table for different feature selection methods, wherein the differences between the selected parameters/metrics are illustrated. The rows in grey are SCADA-dependant parameters. *N.b.* the features ACC FA and ACC SS are the fore-aft and side-to-side accelerations captured by the nacelle's low-quality acceleremoter.

| | dea | dem | $g_2$ | max | mean | median | min | mode | range | rms | $g_1$ | $m_2$ | $m_3$ | $m_4$ | std |
|---|---|---|---|---|---|---|---|---|---|---|---|---|---|---|---|
| ACC BL X | †, △, ∘, *, □ | | | †, △, ∘, □ | | | †, △, ∘, □ | | †, △, ∘, □ | †, △, ∘, *, □ | | | | | †, △, ∘, *, □ |
| ACC BL Y | †, △, ∘, □ | | | †, △, ∘, □ | | | †, △, ∘, □ | | †, △, ∘, □ | †, △, ∘, □ | | | | | †, △, ∘, □ |
| ACC BL FA | ♦, *, †, △, ∘, *, □ | | | †, △, ∘, □ | | | †, △, ∘, □ | | ♦, †, △, ∘, *, □ | †, △, ∘, *, □ | | | | | †, △, ∘, *, □ |
| ACC BL SS | †, △, ∘, *, □ | | | | | | †, △, ∘, □ | | †, △, ∘, □ | †, △, ∘, *, □ | | | | | †, △, ∘, □ |
| ACC BL X disp | †, △, ∘, □ | | | †, △, ∘, □ | | | †, △, ∘, □ | | †, △, ∘, □ | †, △, ∘, □ | | △, □ | | △, □ | †, △, ∘, □ |
| ACC BL Y disp | †, △, ∘, *, □ | | | †, △, ∘, □ | | | †, △, ∘, □ | | †, △, ∘, □ | †, △, ∘, □ | | △, □ | | △, □ | †, △, ∘, □ |
| ACC BL FA disp | ♦, ×, *, †, △, ∘, *, □ | | | ×, †, △, ∘, *, □ | | | ×, †, △, ∘, *, □ | | ♦, ×, *, †, △, ∘, *, □ | ♦, †, △, ∘, *, □ | | ×, †, △, ∘, □ | | ♦, ×, *, △, □ | *, †, △, ∘, *, □ |




| | | | | | | | | | | | | | | |
|---|---|---|---|---|---|---|---|---|---|---|---|---|---|---|
| ACC BL SS disp | †, △, □ | | | †, △, ○, □ | | | †, △, ○, □ | | †, △, ○, □ | △, □ | | △, □ | | △, □ | △, □ |
| ACC ML X | †, △, ○, *, □ | | | †, △, ○, □ | | | ◆, †, △, ○, □ | | ◆, †, △, ○, □ | †, △, ○, *, □ | | | | | †, △, ○, *, □ |
| ACC ML Y | †, △, ○, □ | | | †, △, ○, □ | | | †, △, ○, □ | | †, △, ○, □ | †, △, ○, *, □ | | | | | †, △, ○, *, □ |
| ACC ML FA | †, △, ○, *, □ | | | ◆, †, △, ○, □ | | | †, △, ○, *, □ | | ◆, ⋆, †, △, ○, *, □ | †, △, ○, *, □ | ◆ | | | | †, △, ○, *, □ |
| ACC ML SS | †, △, ○, *, □ | | | †, △, ○, □ | | | †, △, ○, □ | | †, △, ○, □ | †, △, ○, *, □ | | | | | †, △, ○, *, □ |
| ACC ML X disp | †, △, ○, □ | | | †, △, ○, □ | | | †, △, ○, □ | | †, △, ○, □ | †, △, ○, □ | | △, □ | | △, □ | †, △, ○, □ |
| ACC ML Y disp | †, △, ○, □ | | | †, △, ○, □ | | | †, △, ○, □ | | †, △, ○, □ | †, △, ○, □ | | △, □ | | △, □ | †, △, ○, □ |
| ACC ML FA disp | ◆, ×, ⋆, †, △, ○, *, □ | | | ×, †, △, ○, *, □ | | | †, △, ○, *, □ | | †, △, ○, *, □ | †, △, ○, *, □ | | †, △, ○, □ | | ◆, ×, ⋆, △, □ | †, △, ○, *, □ |

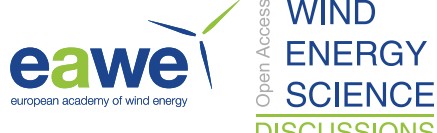

| | | | | | | | | | | | | | | |
|---|---|---|---|---|---|---|---|---|---|---|---|---|---|---|
| ACC ML SS disp | △, □ | | | △, □ | | †, △, □ | | †, △, □ | △, □ | | △, □ | | △, □ | △, □ |
| ACC UL X | †, △, ○, *, □ | | ★ | †, △, ○, □ | | †, △, ○, □ | | †, △, ○, *, □ | †, △, ○, □ | | | | | †, △, ○, □ |
| ACC UL Y | †, △, ○, *, □ | | | †, △, ○, □ | | †, △, □ | | †, △, ○, □ | †, △, ○, *, □ | | | | | †, △, ○, *, □ |
| ACC UL FA | †, △, ○, *, □ | | | †, △, ○, *, □ | | †, △, ○, □ | | †, △, ○, *, □ | ★, †, △, ○, *, □ | | | | | †, △, ○, *, □ |
| ACC UL SS | †, △, ○, □ | | | △, □ | | △, □ | | †, △, ○, □ | †, △, ○, □ | | | | | †, △, ○, □ |
| ACC UL X disp | †, △, ○, □ | | | †, △, ○, □ | | †, △, ○, □ | | †, △, ○, □ | †, △, ○, □ | | △, □ | | △, □ | †, △, ○, □ |
| ACC UL Y disp | †, △, ○, □ | | | †, △, ○, □ | | †, △, ○, □ | | †, △, ○, □ | †, △, ○, □ | | △, □ | | △, □ | †, △, ○, □ |
| ACC UL FA disp | ♦, ×, ★, †, △, ○, *, □ | | | †, △, ○, *, □ | | ♦, ×, ★, †, △, ○, *, □ | | ×, †, △, ○, *, □ | †, △, ○, *, □ | | †, △, ○, □ | | ×, ★, △, □ | †, △, ○, *, □ |
| ACC UL SS disp | △, □ | | | †, △, □ | | †, △, □ | | †, △, □ | △, □ | | △, □ | | △, □ | △, □ |



|  |  |  |  |  |  |  |  |  |  |  |  |  |  |  |  |
|---|---|---|---|---|---|---|---|---|---|---|---|---|---|---|---|
| ACC FA |  |  |  | †, △, □ | †, △, ○, □ | †, △, ○, □ |  |  | †, △, □ | †, △, ○, □ |  | △, □ | △, □ | △, □ | †, △, ○, □ |
| ACC SS |  |  |  | △, □ | △, □ | △, □ |  |  | △, □ | △, □ |  | △, □ | △, □ | △, □ | △, □ |
| Temp |  |  |  |  |  |  |  |  |  |  |  |  |  |  |  |
| Pitch |  |  |  |  |  |  |  |  |  |  |  |  |  |  |  |
| Power |  |  |  | △, □ |  |  |  |  |  |  |  |  |  |  |  |
| Rpm |  |  |  | △, □ |  |  |  |  |  | ⋆ |  |  |  |  | ♦ |
| Wind direction |  |  |  |  |  |  | ♦ |  |  |  |  |  |  |  |  |
| Wind speed |  |  |  | †, △, ○, □ | † |  |  |  | †, △, □ | □ |  | □ |  | △, □ | □ |
| Thrust |  | ♦, ×, ⋆, †, △, ○, □ |  | ×, ⋆ | ♦ |  |  |  | ♦, ⋆ |  |  | ♦, ×, ⋆ |  | ♦, ×, ⋆ | ×, ⋆ |



*Author contributions.*  Conceptualization, Francisco d N Santos, Nymfa Noppe, Wout Weijtjens and Christof Devriendt; Formal analysis, Francisco d N Santos; Funding acquisition, Christof Devriendt; Supervision, Nymfa Noppe, Wout Weijtjens and Christof Devriendt; Writing – original draft, Francisco d N Santos; Writing – review  editing, Francisco d N Santos, Nymfa Noppe, Wout Weijtjens and Christof Devriendt.

All authors have read and agreed to the published version of the manuscript.

*Competing interests.*  The authors declare that they have no conflict of interest.

*Acknowledgements.*  This work was conducted in the framework of the ICON SafeLife : Lifetime prediction and management of fatigue loaded welded steel structures based on structural health monitoring. The authors additionally acknowledge the support of VLAIO.





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
