# Peer review of "Data-driven farm-wide fatigue estimation on jacket foundation OWTs for multiple SHM setups"

_Wind Energy Science, 2021_

## Author Comment (AC1)

**Response to RC1 on Data-driven farm-wide fatigue estimation on jacket foundation OWTs for multiple SHM setups**

-The methodology section should improve a lot. Section 2.2.1 discusses Tier one which is about the estimation and validation of thrust load. Then Section 2.2.2 is focused on feature selection so that it reviews different techniques but does not clearly answer the question that which technique is applied to this work. The reader expects to realize what you have applied in the methodology section. Please provide references for the techniques that you have mentioned there. Maybe it is better to name the Section methodology, then creating two subsections Input data and the Proposed Estimation Algorithm. Please improve the structure to make it easier for readers to follow the work.

The text on the methodology section is indeed confusing. The paragraphs: "The main methodology of the present contribution can be understood as a *two-tier* neural network model. The first tier concerns itself with the generation and processing of relevant 10-minute features, which will serve as inputs for the second tier. In the first tier, an ANN model is utilized to estimate the thrust load on a 1s-basis. This model has to be trained, validated and cross-validated before being considered fit for employment (upper white-background section of Figure 4). After this, the 1s thrust load, along with SCADA and accelerations from the SHM accelerometer, is processed into a variety of 10-minute metrics (*cf.* grey section of Figure 4). Between the 10-minute feature generation tier (grey section of Figure 4) and the DEL prediction tier (pale yellow section of Figure 4), the 10-minute metrics undergo a dimensionality reduction procedure based on a feature selection algorithm, which allows to train, validate and cross-validate a 10-minute ANN FA DEL estimation model for a smaller amount of relevant input features. The motivation behind a 10-minute approach lies with the common framework for data processing (also 10-minute), the aim of including environmental effects and vibration levels, but also the issues inherent to working with different sampling frequencies (1$Hz$ SCADA, 12.5$Hz$ accelerometer) and possible time-delays. After training, validating and cross-validating the DEL ANN model (middle white-background section of Figure 4), we can observe this second tier in pale yellow in Figure 4}. Albeit the current contribution's methodology can be seen globally in Figure 4, the sections which are under a white background consist of training, validation, cross-validation and determining which features ought to be used. Thus, these tasks will only be performed once. When a final model is achieved, only the sections with the grey and pale yellow background are required to produce estimation - this will be picked up again by subsection 3.4." as been added and the author hopes that it is sufficiently enlightening to ensure the continuity of the current structure of section 2.

-Please explain why ADAMAX is used for optimizing ANN training. What are other alternatives and what is making ADAMAX different in this context?

In the manuscript it is explained that ADAMAX is selected through hyperparameter tuning, which is a commonplace term in machine learning to refer to trial-and-error. The choice on an ADAM based optimizer is also explained as being a common practice in the machine learning community: "A common, well-performing choice…". This optimizer was compared to a number of other optimizers available in keras and selected for being the best performing. The sentence: "This particular optimizer was selected through hyperparameter tuning by comparison with other available optimizers, such as Stochastic Gradient Descent (SGD), Root Mean Squared Propagation (RMSProp) and Adadelta (Zeiler, 2012), an extension of the Adaptive Gradient Algorithm, Adagrad." was added to the manuscript.

-Is it possible to provide a confidence interval for the estimation error by applying different techniques like what is used in the following publication: Online condition monitoring of floating wind turbines drivetrain by means of digital twin, FK Moghadam, AR Nejad - Mechanical Systems and Signal Processing, 2022

This suggestion is taken wholeheartedly, but it is the authors' opinion that it would further increase the size of an already big article. Nevertheless, the authors continue their work on this research and will attempt to follow this suggestion in future work.

-Please list the general specification of the under investigation turbines, including the rated power and speed.

The sentence: "These turbines have a rated power of 6 MW and a maximum of 12 RPM." has been added to line 118.

-Could you give a comment on Figures 6 and 13 results, and explain in the text how the error is correlated to wind speed?

The discussion of how the error is correlated with wind speed is present for Figure 6 in: "The increase of MAE with the wind speed is expected, as higher loads are attained for higher wind speeds (and the MAE relates to the absolute value of the loads)." As for Figure 13, it is said: "we can observe how the model performs worse for a different turbine for mid-range speeds (8-18 m/s) rather than for higher speeds. This is possibly related with the worse performance of the model under wake." This comment indeed highlighted a deficiency in the explanation of how the DEL prediction error is correlated with wake, which doesn't show as a linear correlation as for Figure 6. The sentence: "This can be further verified by inspecting Figure 14, where the turbulence intensity is plotted against the wind speed. As turbulence intensity (TI) is given by $TI\ (\%) = (\sigma(u)/\bar{u}) \times 100$), where $\sigma(u)$ represents the standard deviation of wind speed and $\bar{u}$ the mean wind speed, the values of turbulence intensity for lower wind speeds are rather unstable and thus, disregarded in this analysis. In this figure we can see how, for wind speeds between 5 and 18m/s the turbulence intensity of the cross-validation turbine is noticeably higher than for the validation (and training) turbine. We can then reasonably assume that, precisely because the cross-validation turbine faces higher turbulence (being located under more severe wake) than the training turbine, the prediction error for the cross-validation turbine will be highest in the regions were the gap between turbulence intensities is greater". and the following figure, with the caption: "Turbulence intensity (%) *vs.* wind speed (*m/s*) for validation and cross-validation turbines." were added.

[Figure]

-The highest error in DEL does not correspond to the highest error in thrust load estimation. Could you add a comment on that? Is it possible to do a sensitivity analysis to see which parameter contributes the most to the DEL error?

Thrust load and total DEL don't have a 1:1 relation – the computation of total DEL, although dependent of the thrust load, is also the fruit of complex interactions between thrust, SCADA and accelerations. It should also be noted that Fig. 6 and Fig. 13 are comparing two different metrics: Fig. 6 presents the MAE (in % of maximal training thrust) computed at every 10 minutes and Fig. 13 has the Error (as a percentage of the maximal training DEL). The suggestion of performing a sensitivity analysis to see the contribution each parameter has on the DEL error is indeed much appreciated and taken wholeheartedly, but the author feels that the current manuscript is already rather extensive. Nevertheless, the research on this topic is ongoing and this suggestion will surely be included in future work.

-Could you move the schematic diagram presented in Fig. 15 related to the farm-wide DEL to the methodology section?

The author hopes that the discussion of the first comment regarding the methodology makes this suggestion unnecessary. The sentence: "Figure 16 schematizes the pipeline for the farm-wide model, consisting in the parts of Figure 15 which are kept for producing DEL estimations - the 10-minute feature generation and DEL predictions tiers, in grey and pale yellow, respectively". has been added to page 24 line 548.

-Line 125 "To avoid excessive drifts in this transformation a lower frequency bound of 0.1Hz is used.", please elaborate more or possibly provide a reference.

It is explained in "Dynamic strain estimation for fatigue assessment of an offshore monopile wind turbine using filtering and modal expansion algorithms" how, for low frequency strains, induced by the thrust-loading there's a concern with the conversion from accelerations to strains which requires a double integration, risking blowing up the low frequency noise in the measured accelerations, resulting in large errors in the low frequency components of the predicted strains, requiring a high pass filter to prevent this (see Section 3.5, page 15-17, K. Maes et al., Dynamic strain estimation for fatigue assessment of an offshore monopile wind turbine using filtering and modal expansion algorithms, Mech. Syst. Signal Process. (2016)). This reference has been added to the manuscript.

-Line 152 "In the current contribution we focus primarily on the DEL estimation in the FA direction, as this is considered most relevant for the current jacket foundations." Could you provide a reference?

The sentence was altered to: "In the current contribution we focus primarily on the DEL estimation in the FA direction, as measurements reveal it to be the most relevant for the current jacket foundations."

-Line 170 "The thrust load can be obtained from measurements by low-pass filtering the bending moment timeseries (with an upper frequency bound of 0.2 Hz)." Why is the cut-off frequency chosen to be 0.2 Hz?

It is explained in "Performance monitoring and lifetime assessment of offshore wind farms based on SCADA data" how up until 0.2 Hz the frequency spectrum can be interpreted has being in a quasi-stactic (thrust load) region. Above 0.2 Hz begins the low frequency region driven by wave dynamics and the first eigenfrequency. (See Fig. 5.5., page 112, N. Noppe, Performance monitoring and lifetime

assessment of offshore wind farms based on SCADA data). This reference has been added to the manuscript.

-Site-to-side in figure 2 should change to side-to-side.

Altered.

-Multiple grammatical errors. e.g. "a intermediate" in Line 27 that should be corrected.

The manuscript was further proof-read, and errors corrected.

---

## Author Comment (AC2)

**Response to RC2 on Data-driven farm-wide fatigue estimation on jacket foundation OWTs for multiple SHM setups**

*Comments:*

At the top of Figure 4, I read the expression "Measured thrust load". I would not agree that thrust is strictly measured, it is rather estimated from the strain measurements.

This is correct. The sentence: "*N.b for ease of writing, the thrust load deduced from the strain gauges is mentioned as measured thrust load, and to differentiate from the ANN's thrust load predictions.*" Was added to the caption of Figure 4.

At the top of Figure 4, I read that you provide a validation set for the thrust load. This indicates that the thrust has been validated independently somehow. Might you please elaborate on this point? Were validation performed via an aeroelastic simulation? In order to get a good estimate of thrust, the strain gauges need to be calibrated for a know load level. How was this done, in order for one to trust that the Thrust estimates are correct?

The aforementioned figure was indeed problematic, as it led the read to believe that the thrust load was being validated. In fact, the terms 'validation' and 'training' of Figure 4 relate to the validation and training of the thrust load ANN model – this has been corrected in the figure to show 'Training & Validation' for both the 1s SCADA and thrust load.

On page 9: I suggest that authors explain the basis by which they selected the features of the time series.

The reason which features were included is related with Vera-Tudela's study. In the text (line 220) it is highlighted that Vera-Tudela's paper signals spectral moments, skewness and kurtosis has features of interest, as they aren't that commonly used, but the other features are also present in this reference. This sentence has been altered to: "The selection of which engineering input features should be calculated can be traced to Vera-Tudela and Kühn (2014)".

Section 2.2.2: another powerful feature selection approach is https://pypi.org/project/BorutaShap/

This is indeed a good suggestion and, has the current research of the group will again involve feature selection, it will surely be added to the list of techniques.

Page 12 line 289-290: "carefully selected as to be representative of all operating conditions". What are the criteria for such a selection? Turbine operating in partial load, full load and transition? Turbine operating at max Cp and rated power? Operating conditions covering all possible combination of pitch-tip Speed Ratio? Please explain what are all the representative operating conditions.

The sentence: "…, namely parked, run-up and full load" was added in line 302.

Page 12 line 292: why do you correct the thrust for the air density? Are you calculating the thrust coefficient?

The answer to this question can be found in Noppe, N., Weijtjens, W., and Devriendt, C.: Modeling of quasi-static thrust load of wind turbines based on 1 s SCADA data, Wind EnergyScience, 3, 139–147, 2018: "According to Baudisch (2012), thrust loads are influenced by air density. While changes in the depicted SCADA variables happen within seconds, air density changes on a different timescale (several hours). Instead of including air density in the set of input parameters, it is accounted for as a correction of the modeled thrust load $\widehat{F_T}$: $\widehat{F_{T,corr}} = \rho\, \widehat{F_T}$". Both references have been added to that section of text.

Page 13 line 299: ANN as used in this article cannot extrapolate, i.e. they cannot make correct predictions when the input are not within the range of the training set. How do the authors ensure that the validation data (3 months of data outside of the training period) fall within the range of the training set?

The training data was ensured to be statistically representative of all conditions faced during 1-year worth of data. This is further discussed in d N Santos, F., Noppe, N., Weijtjens, W., and Devriendt, C.: Input parameter selection for full load damage neural network model on offshore wind structures, in: Proceedings of 16th EAWE PhD Seminar on Wind Energy, 2020 for the DEL model. The sentence: "… carefully selected as to be statistically representative of all operating conditions …" has been added at line 301.

Page 13 line 300: the cross-validation applied to a different turbine. Has the cross-validation set been chosen in such a way to reflect the conditions that occurred in the original training set? i.e. you cannot cross-validate on a set where the other turbine is known to be in a waked condition, right?

In relation to wake, even though the training turbine is mostly under free flow, it also includes training periods in which it is under wake. The training turbine is mostly free-flow facing, but can be understood as under partial wake. The assumption was that the present of partial wake would be taken into account by the model and suffice – apparently this assumption doesn't hold.

Table 1: you want to consider adding this to the mix of methods:
https://pypi.org/project/BorutaShap/

Answered above.

Page 15: please note that random forest feature importance can be derived either based on mean decrease in impurity or based on feature permutation. Both methods could give slightly different results. Mind you that the results could also be affected by dependent input features. Please make sure you do not bias your feature selection based on the above. Kendall's Tau takes such dependence into account, and probably why it gives the lowest MAE in figure 7 at the cost of high number of features

The author acknowledges the reviewer's comment and will keep it in mind for future research. In the present contribution it was only used for comparison and not for the final results.

Section 3.2.2: Table 2: once 1-second SCADA data is made available, by default then the 10min-SCADA can be calculated. Did you consider a scenario where both 10min and 1Hz SCADA are made available to the model?

The 10-minute mean of 1s SCADA is the same as 10-minute data. What changes is simply the availability of features (with 1s SCADA, apart from mean, you can have spectral moments, etc.)

Section 3.2.2:  it is perfectly acceptable to propose several scenarios of various data sources, and check their effect on the model predictive error. However, wouldn't a more principled approach involving sensitivity analysis and sensors selection optimization with value of information be more adequate? Please discuss.

One of the driving research questions behind this contribution was to understand the models' performance degradation when accelerations aren't included in the model. It was also driven by a desire to understand what some real-world sensor setups would allow to achieve – a goal that is mirrored by a common operator concern, namely, is it worth installing X or Y sensor, and what am I gaining/missing out with each installation. The author isn't entirely acquainted with value of information analysis, but from its understanding value of information requires a cost-benefit analysis on sensor uncertainty, which is currently out of scope. Nevertheless, the author will begin shortly collaboration with an external partner aiming precisely at quantifying the uncertainty of these models and the gain in reducing those uncertainties.

Page 20, line 472: please elaborate on the 80-20 train-test split. Do you respect the temporal evolution of the data or do you perform randomized split?

The train-test split was randomized (this information has been added to line 486).

Page 21, line 489: "…for a different turbine". Please specify where this other turbine located with respect to the reference turbine where the model was trained.

Added: " – located in the turbines' wake –" and "The training turbine is located at the north-western edge of the farm and the cross-validation turbine located in the middle of the farm (*cf.* subsection 3.4., OWT 7 and OWT 35 Figure 17)" at lines 489-491.

Page 22, line 505-508: this might the case, but the question is whether the training set of the reference turbine included any data corresponding to wake? Indeed not all wakes/partial wakes/multiple wakes are created equal because of dependence on atmospheric stability, turbulence and shear. It is worth discussing this issue. The logical consequence of this is that your training set for the reference turbine should include a much larger amount of data in order to take into account the various wake effects in order to generalize to another turbine…

This is indeed true, albeit the training turbine is under partial wake. For reference, the training turbine was OWT 7 and the cross-validation turbine OWT 31. Naturally that, the larger the training dataset, the better the training turbine will be able to adapt and generalize. One of the questions that arise is whether the use of a transfer function is able to bridge this higher need of data. Or a population-based approach, for which a future approach is planned.

Page 22, line 505-508: 8-18 m/s according to the reference turbine where the model was trained or according to the different (second) turbine? If the second turbine were wake affected, then its wind speed should account for the velocity deficit.

The binned wind speeds of the validation turbine are of this turbine. The same for the cross-validation turbine.

Page 22, line 505-508:   Avendaño-Valencia, L. D., Abdallah, I., and Chatzi, E.

The sentence "Avendaño-Valencia et al. have worked in this direction, concluding that the fatigue life of OWTs under free-stream inflow can be quite distinct from OWTs under wake (Avendaño-Valencia et al.)" has been added to lines 523-526.

Figure 16: what is mean by mean_DEL? DEL is a short term measure of fatigue conditional on wind speed. Is the mean calculated by weighting according to the pdf of the wind speed? Please elaborate how the mean is computed in this case.

The mean here represents the simple arithmetic mean of every 10-minute DELs. Added "(arithmetic mean of all 10-minute DELS)" to caption

Figure 18: same comment as above.

Same as above.

Figure 16: DEL across various wind turbines in the farm will be highly influenced by the mean wind speed (at each wind turbine), turbulence and shear. Maybe you would want to plot Figure 16 for various wind speed bins (e.g. 6-9m/s, 10-14m/s and >15m/s). You will notice that the DEL across wake-free and wake affected wind turbines will change quite a bit.

This is indeed a good point, but it would add yet another topic to an already rather lengthy paper and the main objective of subsection 3.4 was to merely present a farm-wide application of the contribution's methodology, without going too much into detail. Nevertheless, it is in the researcher's plans to dedicate a proper study to farm-wide wake.

*General comment:*

There are multiple grammatical and orthographic mistakes, and a proof-read is necessary.

Revised

It is not mandatory, but you might want to consider comparing the performance of your method to other state of the art methods dealing with a similar subject matter.

This comparative aspect has been briefly explored in the introduction section (1.2), from a methodological POV.

*Methodological Suggestion:*

A more direct approach avoiding the two-tier approach proposed in this article might be to use variational auto-encoders neural networks such as proposed here:

https://onlinelibrary.wiley.com/doi/full/10.1002/we.2621

https://link.springer.com/chapter/10.1007%2F978-3-030-12075-7_21

These references have been added to the introduction section (line 104): "As for *Mylonas et al.*, it used conditional variational auto-encoder neural networks to estimate the probability distribution of the accumulated fatigue on the root cross-section of a simulated wind turbine blade, making long-term probabilistic deterioration predictions based on historic SCADA data (*Mylonas et al.,2020, 2021*)".

---

## Author Comment (AC3)

**Response to RC3 on Data-driven farm-wide fatigue estimation on jacket foundation OWTs for multiple SHM setups**

General comments:

There are some typing and/or grammatical errors that should be revised. For example, 'within the an offshore wind farm' in Line 114-115.

The manuscript was further proof-read.

One of the unique aspects of this work is stated to be the use of ANN estimated thrust load as an input for the fatigue estimator (Line 72). It would then be sensible to compare the effect of including or excluding the thrust load on the accuracy of the fatigue estimation. Only one such comparison is possible in Section 3.2.2 between scenario D and scenario F. The cross-validation for these scenarios however show that the inclusion of the thrust load increases the MAE which may result due to overfitting. This is briefly mentioned in Line 441-444, but for the farm wide evaluation (Section 4.3) the thrust load is included even though it has been shown to result in overfitting. It is recommended that the authors evaluate and discuss whether the use of estimated thrust load is beneficial to a model that can be used for farm-wide estimates or whether scenario D would not better address the goals of this work.

In addition, the impact of including/excluding thrust load is said to be studied in future work (Line 581). If the inclusion of the thrust load is indeed deemed to be unique to this work, it falls within the scope of this paper and should be addressed thoroughly.

This comment is much appreciated and has indeed made a very fair point which led to re-writing the entirety of section 3.4. The authors invite the reviewer to revisit this section. But shortly, it can be said that both scenarios (D and F) were included, as one presents better values for validation and the other for cross-validation. The overall trends are the same for both plots, apart from one outlying behaviour – which is further explained.

Comments:

Line 118: Rotational speed is also referred to as RPM which is confusing as the unit in which rotation is measured is cpm and not RPM. Recommended use of only rotational speed.

Correct to just present rotational speed

Line 119: Unit of ambient temperature is given as ° which does not distinguish between °C and °F.

Corrected to °C

Line 368: This is the first time that reference is made to the X direction in the text. At that stage it was unclear what the X direction is. It was found that Figure 2 also refers to X and Y, but it might be clearer if the text mentions that X and Y is the measurement directions of the accelerometers (perhaps in Line 121-122). Possibly also show in a figure like Figure 16 how these directions are defined.

Added X and Y as the sensor directions in line 122: "Apart from the accelerations in the two sensor directions, X and Y…".

Line 388: Y vibration is said to be unnecessary. Is this due to yaw angles close to 0° or 180° for most data used during feature selection? If the yaw angle is close to 90° or 270°, the FA direction would be the same as the Y direction. Would this case not result in the Y SHM measurements to be better correlated to the fatigue? Perhaps the exclusion of Y measurements would not generalize for all conditions of the studied turbines or generalize well for all wind farms.

This is indeed correct, but feature selection should always be redone for different sites, and conclusions related with selected features are relevant for the current site. The sentence: "Apart from the accelerations in the two sensor directions, X and Y (with X in the dominant wind direction, often aligning with the fore-aft and Y 90° from it)" was added at lines 126-128. The sentence: "SS and Y SHM accelerations appear to be unnecessary for this site" was added at lines 400-401. The sentence: "The authors would like to point that the conclusions related to which features are to be selected is connected with the site at question and that it is a good practice to redo feature selection for a different site" was added to lines 421-423.

Line 424: The final scenario is that of the final instrumentation setup with only the selected features used as inputs for the ANN. It is recommended that an additional scenario is included in this section which uses the same instrumentation but no feature selection. This would be beneficial to see the impact of the feature selection on the errors for validation and cross-validation. This would then show the increased performance due to feature selection which is referred to in Line 229.

This suggestion is taken wholeheartedly, but the author is of the opinion that it isn't fully necessary. The main reason for the feature selection isn't just connected with improving the models' performance (which is, admittedly, stated in section 2.2.2.), but mainly with helping identify key features/input parameters and performing dimensionality reduction. Indeed, with over 430 engineered features, the training of models and processing of such an amount data would result in a very time-consuming process.

Line 451: The use of the word 'clearly' seems to contradict the initial understanding of Figure 10. This is evidenced by the need of the author to explain the lack of generalization in Line 452. If one were to choose a case that is generally 'best' based on Figure 10, it seems that scenario G performs better. Please explain how these results are used to determine the superiority of scenario H.

The use of the word 'clearly' is indeed abusive and contradicts the spirit of the text. The issue with Figure 10, and scenarios G and H, is that, albeit G presents a slightly better MAE cross-validation value, H has a better result for the validation turbine. Additionally, we can observe that the spread for both the box and whiskers of H is lower than G. If we compare the error (%) values for models H and G, we can see that for both cases the boxplots for model H have a lower spread for most wind speeds (including for cross-validation). The MAE might be marginally lower for G because it's general offset might be lower than for H. In sum, MAE is a valuable metric, but, as a single value, it can't show the entire picture. Altered sentence to: "Finally, the results for the model based on the RFE DTC selected features are the best for the turbine (H), as it presents the lower spread in its boxplots."

[Figure]

Line 471: Is the 8000 datapoints each selected randomly or is a period of 8000 consecutive datapoints selected randomly?

8000 randomly selected datapoints (non-consecutive). Added 'non-consecutive' at the sentence referenced.

---

## Author Response (AR2)

**Response to 2nd round on Data-driven farm-wide fatigue estimation on jacket foundation OWTs for multiple SHM setups**

The authors would like to express their acknowledgement to the reviewer and editor and thank them for their work.

Editor comments:

The authors meaningfully engaged with the reviewer comments by adding clarifying statements to the text. There are no apparent changes to the scientific content of the document.

A rewriting of section 3.4 aimed at clarifying the apparent lack of comparison between cases that include and exclude the estimated thrust load. This comment stemmed from the authors identifying the inclusion of the estimated thrust as the distinguishing component of the research while not seeming to thoroughly address whether this inclusion adds to the accuracy of predictions.

The rewriting of section 3.4 contains the two best options (one which uses the estimated thrust and one that does not) for farm-wide DEL estimation. Even though reference is made to both options, the authors offer no discussion comparing the superior of these two options. This leads the reader to use the comments from section 3.2.2 that the inclusion of thrust results in overfitting of the model which degrades the generalisation of the model to other turbines (Line 459-460). This apparently shows that the inclusion of thrust is not a good idea if a fleet leader concept is to be used.

The aim of this paper is stated to "assess the feasibility of this strategy, study the added value of various sensors and statistics and provide insight in how the most suitable parameters can be selected" (Line 76-77). This aim is especially relevant for the ANN estimated thrust load which is unique to this work (Line 72).

The authors however do not refer to whether this "unique" aspect was indeed beneficial to the creation of a farm-wide DEL estimator. From the presented results, it appears that the inclusion of the thrust load decreases the accuracy of the model when using it for farm-wide applications. The only reference to the importance of the thrust is not substantiated with external references or the results of this work ("It should, however, be said that, in itself, the thrust load is a highly relevant parameter, be it standalone, or if intended for further use in training.

It is recommended that the authors critically evaluate what the impact is of including the thrust load in the accuracy of a farm-wide model from the presented results and remove references to future work that will evaluate this impact.

It is the opinion of the reviewer that this would increase the clarity of this work. This comment was made during the first round of revisions after which the authors made changes to the document which did not completely clarify all objections of the reviewer. It is requested that the editor evaluate the validity of this comment and communicate with the authors if further changes is deemed to be necessary.

This is indeed a fair comment. And we acknowledge that the answer on whether to include the thrust load or not is not fully given in this work. However, in our opinion this was also not the key learning objective as we also look into the added value of e.g. 1s SCADA in general and a (high quality) accelerometer, a comparison of different feature selection strategies and a study on the feasibility of ANN for fatigue estimations of offshore wind turbines considering different restrictions in available data.

While initial working version(s) of the manuscript started by considering the thrust load model as a default part of the methodology (and perhaps this was still somewhat reflected in some passages), we now consider it as just another possible source of data. The text has been reworked (in particular the abstract) to reflect the reduced role of the quasi-static thrust load and avoid any remaining confusion.

To answer the question whether the thrust load model should be included we're currently restricted by the considered data. The limiting factor in this matter is that at this farm we only have two turbines instrumented that allow us to the validate the models. In the current research we opted to consider one turbine as our training turbine and one turbine for cross-validation. Or, in other words, we imposed ourselves the restriction that the training data would originate from just one turbine.
From the validation data we learn that the inclusion of the quasi-static load model does offer a clear improvement over all other models. However, from the cross-validation we now learn that this might translate in a slightly worse transferability of the model and we openly discuss so.
Arguably we could train our thrust load model on both turbines and likely this would result in a better performance towards cross-validation at a reduced validation cost. But in absence of more instrumented turbines we have no way to validate this model outside the training turbines. It is for this reason we opted to keep the discussion as is, concluding that the inclusion of the thrust model offers the best validation score, however we must be aware of the lower cross-validation score. We've revised the text to assure this conclusion is consistent throughout the manuscript. We also put this more explicit in the manuscript's conclusion. We also make no claims that the thrust model is essential to the success of the algorithm.

Reviewer comments:

Figure 16: The colour scale for the side-by-side figures differs. This reduces the ability to compare the results generated from scenario D and scenario F. A consistent colour scale would be beneficial to the interpretation of the figure.

This comment is indeed valid. As such, figures 16(a) and 16 (b) have been altered to have a common scale and the colour map changed to better allow to identify outliers within each plot. It ought to be signalled that, albeit upon first glance the plots seem radically different in their values, this is due to the narrowness of the scale. Taking the example of OWT12: for the first plot it has a mean DEL of about 0.18, whereas for the second plot this value is 0.24. One could interpret this as being a difference of over 30% (0.24-0.18/0.18). However, as the mean DEL is already expressed as a ratio of the maximal training DEL value (in this case, and due to the considered period being summer months, the ratio will be rather small) the difference between both models would be simply 0.24-0.18=0.06 (or 6%). This is also reflected in Figure 10, wherein the difference of the mean between models D and F is about 4%.

[Figure]

Figure 16(a)

[Figure]

Figure 16(b)

[Figure]

Figure 10

Line 574, Figure 18: Reference to the mean acceleration. Mean acceleration is usually zero as it oscillates around zero. Please clarify if this parameter is indeed the one used for plotting Figure 18.

The mean acceleration does indeed usually oscillate around zero – this is the case of the high-quality SHM accelerometers installed at three points of the tower. However, as the farm-wide implementation requires the use of low-quality accelerometers installed at nacelle level (present for all sites), this doesn't hold true, as can be attested by Figure 3 (green line). Indeed, these sensors present the absolute value of accelerations. In order to avoid confusions, the reference to absolute accelerations was added whenever 'low-quality accelerations' are mentioned in the text and captions.